# Rescaled Influence Functions: Accurate Data Attribution in High Dimension

**Ittai Rubinstein**
EECS and CSAIL
MIT
Cambridge, MA
ittair@mit.edu

**Samuel B. Hopkins**
EECS and CSAIL
MIT
Cambridge, MA
samhop@mit.edu

## Abstract

How does the training data affect a model's behavior? This is the question we seek to answer with *data attribution*. The leading practical approaches to data attribution are based on *influence functions* (IF). IFs utilize a first-order Taylor approximation to efficiently predict the effect of removing a set of samples from the training set without retraining the model, and are used in a wide variety of machine learning applications. However, especially in the high-dimensional regime (# params $\geq \Omega$(# samples)), they are often imprecise and tend to underestimate the effect of sample removals, even for simple models such as logistic regression. We present *rescaled influence functions* (RIF), a tool for data attribution which can be used as a drop-in replacement for influence functions, with little computational overhead but significant improvement in accuracy. We compare IF and RIF on a range of real-world datasets, showing that RIFs offer significantly better predictions in practice, and present a theoretical analysis explaining this improvement. Finally, we present a simple class of data poisoning attacks that would fool IF-based detections but would be detected by RIF.

## 1 Introduction

*Data attribution* aims to explain the behavior of a machine learning model in terms of its training data. If $\theta$ is a model trained on a dataset $\{(x_i, y_i)\}_{i \in [n]}$, the fundamental algorithmic task in data attribution is to answer the question:

> ***Leave-$T$-Out Effect:*** *How would $\theta$ have been different if some subset $T \subseteq [n]$ of the training set had been missing?*

The ability to quickly and accurately predict a leave-$T$-out (LTO) effect, or to search for subsets producing a large leave-out effect, unlocks extensive capabilities from classical statistical inference to modern machine learning. For example, the jackknife, leave-$k$-out cross-validation, and bootstrap are all widely used to quantify uncertainty and estimate generalization error or confidence intervals, and all rely on the ability to quickly estimate LTO effects [Efr92, GSL+19, Jae72]. Machine learning has seen an explosion of applications of data attribution, for dataset curation [KL17, KATL19], explainability [KATL19, GBA+23], crafting and detection of data poisoning attacks [EIC+25, KSL22, SS19], machine unlearning [SAKS21, GGHVDM19, IASCZ21], credit attribution [JDW+19, GZ19], bias detection [BAHAZ19], and more.

Ascertaining the ground truth leave-$T$-out effect in general requires a full retrain of a model for each $T$ of interest, which is computationally intractable in all but the simplest settings. Consequently, approximations to the leave-$T$-out effect are widely used. Key desiderata for such approximations

39th Conference on Neural Information Processing Systems (NeurIPS 2025).

are (1) *accuracy*, (2) *computational efficiency* even for large-scale models, and (3) *additivity*: the predicted effect of removing $T$ should be the sum of predicted effects of removing each element of $T$ individually. Additivity enables another important capability: *search* for the subset $T$ of a given size with the greatest predicted effect according to a given metric, by taking the $k$ training data points with largest predicted leave-one-out (LOO) effects [BGM20, IPE$^+$22, HBN$^+$24].

*Influence functions* (IF) [Ham74] are by far the most widely used and studied data attribution method. The IF is a first-order approximation to the change in model parameters when infinitesimally down-weighting an individual sample. IF approximations are well studied in classical, under-parameterized settings, where they are typically accurate and enjoy solid theoretical foundations [GSL$^+$19]. But, despite widespread adoption for data attribution in high-dimensional/overparameterized models, IF's accuracy in the high-dimensional setting is comparatively poor. Empirical studies show that IFs often underestimate the true magnitude of parameter changes, leading to potentially misleading conclusions about data importance or model robustness [BPF21, KL17]. And, existing theoretical analyses justifying IF approximations break down for overparametrized models. But, thus far, more accurate alternatives to IFs have proved too computationally expensive to be practical.

We study a simple and fast-to-compute modification of the influence function, which we term the *rescaled influence function* (RIF). RIFs improve accuracy by incorporating a limited amount of higher-order information about the change in model parameters from sample removal, but retain the additivity and in many settings also the computational efficiency of IFs. We show via experiments and theoretical analysis that RIFs are accurate for data attribution in overparameterized models where IFs struggle. Like IFs, RIFs are model and task agnostic, meaning that they can be applied to any empirical risk minimization-based training method with smooth losses, and they can estimate the leave-$T$-out effect according to any (smooth) measure of change to model parameters. We therefore advocate using RIFs as a drop-in replacement for IFs across data attribution applications.

**Organization**   In Section 1.1, we introduce RIFs formally. Section 2 presents our experimental results, and Section 3 presents our theoretical analysis of RIF. We discuss context and conclusions in Sections 4 and 5

## 1.1   Influence Functions, Newton Steps, and Rescaled Influence Functions

We now introduce the rescaled influence function formally. Suppose that $\{(x_i, y_i)\}_{i \in [n]}$ is a training data set, $\Theta \subseteq \mathbb{R}^d$ is a class of models, and $\ell(x, y, \theta)$ is a twice-differentiable loss function; $\ell$ may include a regularizer. For simplicity, we imagine that $\ell$ is convex, although the definition of RIFs can be extended to the non-convex case. Let $\hat{\theta} = \arg\min_{\theta \in \Theta} \sum_{i \leq n} \ell(x_i, y_i, \theta)$ be the empirical loss minimizer (or, in the non-convex setting, any local minimum of the empirical loss).

**Influence Functions**   The influence function $\text{IF}_i \in \mathbb{R}^d$ associated to the $i$-th training sample is a first-order estimate of the effect of dropping that sample.[1] Introducing a weight $w_i \in [0, 1]$ associated to each sample $i$ and allowing $\hat{\theta}$ to depend on $w$ via $\hat{\theta}(w) = \arg\min_{\theta \in \Theta} \sum_{i \leq n} w_i \cdot \ell(x_i, y_i, \theta)$,

$$\text{IF}_i = - \left[ \frac{d}{dw_i} \cdot \hat{\theta}(w) \right] \Big|_{w=\mathbf{1}} = H^{-1} \cdot \nabla \ell(x_i, y_i, \hat{\theta}) .$$

Here, $H$ is the Hessian of $\sum_{i \leq n} \ell(x_i, y_i, \theta)$ evaluated at $\hat{\theta}$ (see e.g., [RHRS86] for a derivation). For $T \subseteq [n]$, the IF estimate of the leave-$T$-out model is

$$\hat{\theta}_{\text{IF}, T} = \hat{\theta} + \sum_{i \in T} \text{IF}_i .$$

We can obtain all the single-sample IF estimates $\text{IF}_i$ at the cost of a single Hessian inversion and $n$ gradient computations, which then suffice to obtain $\hat{\theta}_{\text{IF}, T}$ for any $T$ via additivity.

**Newton Steps**   IFs are additive and efficiently computable, but their accuracy suffers when $n$ and $d$ are comparable, or, worse still, if $d$ significantly exceeds $n$ as in the overparameterized setting

---

[1]Some treatments replace dropping with up-weighting, with a resulting difference of sign compared to our convention.

([KATL19]; see also Section 2). A much more accurate approximation to the leave-$T$-out effect is given by taking a single Newton step (NS) to optimize the leave-$T$-out loss $\sum_{i \notin T} \ell(x_i, y_i, \theta)$, starting from $\hat{\theta}$. The NS approximation to the leave-$T$-out effect is given by

$$\hat{\theta}_{\text{NS},T} = \hat{\theta} - H_{[n]\setminus T}^{-1} \left( \sum_{i \notin T} \nabla\ell(x_i, y_i, \hat{\theta}) \right) = \hat{\theta} + H_{[n]\setminus T}^{-1} \left( \sum_{i \in T} \nabla\ell(x_i, y_i, \hat{\theta}) \right) .$$

Here, $H_{[n]\setminus T}$ is the Hessian of the leave-$T$-out loss, evaluated at $\hat{\theta}$, and the second equality follows from the fact that $\theta$ is a local optimum of $\ell$.

As early as 1981, Pregibon [Pre81] observes in the context of leave-one-out estimation for logistic regression that the Newton step approximation is remarkably accurate. At a high level this is because, unlike the IF approximation, the NS approximation takes into account the change to the Hessian from removing the samples in $T$. For convex losses, the true leave-$T$-out effect can often be obtained by Newton iteration – taking multiple Newton steps initialized with $\hat{\theta}$. The only differences we expect to see between the one-step NS approximation and the result of Newton iteration would arise because the Hessian may change from its value at $\hat{\theta}$. Thus, for problems with Lipschitz Hessians, we expect NS to be a very accurate approximation to the true leave-$T$-out effect; [KATL19] offers experimental validation of this idea for leave-$k$-out estimation in logistic regression, and some formal justification.

**Rescaled Influence Functions**    The accuracy of the NS approximation comes at significant cost, since each fresh $T$ requires a Hessian inversion, and additivity is lost. The RIF recovers additivity and much of the computational efficiency of IF, but retains much of the accuracy of the NS approximation. For sample $i \in [n]$, let $\text{RIF}_i$ be the NS approximation to the leave-$i$-out effect, given by $\text{RIF}_i = H_{[n]\setminus\{i\}}^{-1} \cdot \nabla\ell_i(x_i, y_i, \hat{\theta})$. Then for $T \subseteq [n]$, we define the RIF approximation to the leave-$T$-out effect to be

$$\hat{\theta}_{\text{RIF},T} = \hat{\theta} + \sum_{i \in T} \text{RIF}_i .$$

RIF is additive by definition.

The computational overhead of RIF compared to IF depends in general on the cost of computing the $n$ leave-one-out Hessian inversions – once these are obtained, no fresh Hessian inversion is needed to compute $\hat{\theta}_{\text{RIF},T}$ for any $T$. RIF is especially attractive in generalized linear models and neural networks with a ReLU activation function, where $\text{RIF}_i$ can be obtained from $\text{IF}_i$ by multiplying by a rescale factor $(1 - h_i)^{-1}$, where $h_i$ is a (generalized) leverage score associated to the $i$-th sample, which can be computed via a single matrix-vector product with $H^{-1}$. Thus, for generalized linear models, no additional Hessian inversion is needed. For example, in logistic regression, the formula for $\text{RIF}_i$ uses the rescaling $(1 - h_i)^{-1}$, where $h_i = \hat{y}_i(1 - \hat{y}_i) \cdot x_i^\top H^{-1} x_i$; here $\hat{y}_i \in [0, 1]$ is the logistic predicted label of the $i$-th sample according to $\hat{\theta}$.

Beyond generalized linear models and ReLU neural networks, whenever each sample makes a low-rank contribution the Hessian, the $n$ leave-one-out Hessian inversions can be computed quickly via the Sherman-Morrison/Woodbury formula. In all of our experiments, the running time overhead to compute RIF is negligible (see Table 2).

In underparameterized settings, it is reasonable to expect that removing a single sample has a negligible effect on the Hessian, and so $\text{IF}_i \approx \text{RIF}_i$. But for high-dimensional or overparameterized models, a single sample removal can have a significant effect on the Hessian. Our experiments and theory demonstrate the significant accuracy improvement of RIF compared to IF in high-dimensional and overparameterized models.

We note that the idea of summing over estimates of leave-one-out effects to estimate the leave-$T$-out effect is not new, and has been a central component of many previous data models [IPE+22]. In their seminal TRAK paper, Park et al. separately consider both the idea of combining LOO effects additively [PGI+23a][Definition 2.3] and the idea of using a Newton step to estimate LOO effects of a logistic regression [PGI+23a][Definition 3.1] but do not explicitly combine the rescaling effect in their estimator except to note that the rescaling correction has little to no effect in their setting.

A similar approach that has been the focus of recent research is the Additive-One-Exact data model, which estimates the LTO effect by summing over the *exact* LOO effects. This data model was

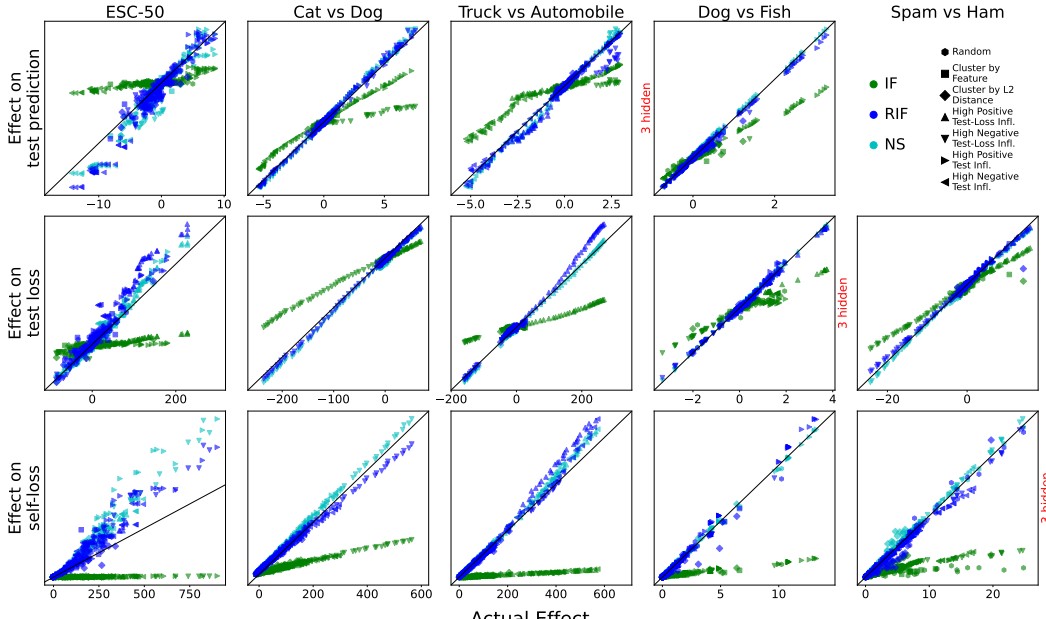

Figure 1: Accuracy of IF versus RIF compared across datasets from image classification (DogFish, Cat vs Dog, Truck vs Automobile), natural language (Spam vs Ham), and audio (ESC-50). In each dataset, we study a binary classification task solved via logistic regression with frozen-embedding features. Each point represents a single choice of subset $T$. The horizontal axis represents ground truth leave-$T$-out effect as measured by changes to test predictions, test losses, and self-loss, computed via refitting the logistic model. The vertical axis represents the prediction of this effect made by IF/RIF/NS. A perfectly accurate prediction falls along the black diagonal line. In essentially every case, the RIF prediction falls nicely along this "ground truth" line, agreeing with the NS prediction, while IF typically underestimates the leave-$T$-out effect.

introduced by Kuschnig et al. [KZCC21] and further analyzed by Hu et al. [HHZM24] and by Huang et al. [HBN+24]. Kuschnig et al., Hu et al. and Huang et al. study the accuracy of this method for identifying sets of highly influential samples in ordinary least squares (OLS) regressions. Moreover, Huang et al. also note that because a single Newton step is equivalent to a full retrain for the case of OLS, a natural extension of the Additive-One-Exact data model is to sum over the single-Newton step attributions of the individual samples [HBN+24][Appendix C.2], and Hu et al. [HHZM24][Section 3.1] hypothesize that an NS-like rescaling might explain some of the inaccuracy of the IF estimates in Koh et al.'s experiments. However, the experiments and theoretical analyses of these previous works focus on the case of OLS linear regression where RIF is equivalent to Additive-One-Exact and to the best of our knowledge, no prior work offers quantitative experimental or theoretical comparisons between RIF and other data attribution methods in the high-dimensional beyond this setting.[2]

## 2 Empirical Results

We now present empirical findings on the accuracy of RIF estimates for leave-$T$-out effects. Our experimental setup is inspired by the seminal work of [KL17, KATL19], who assess the accuracy of influence function estimates using logistic regression as a testbed.

We compare IF, NS, and RIF estimates across the first five datasets in Table 1, spanning vision, NLP, and audio classification tasks. Each dataset is processed using a domain-specific embedding, and we train a logistic regression model to solve a binary classification task on the embedded data. We compare the actual vs predicted effect of removing a given set of samples $T$ from the training set, while varying:

---

[2]We are grateful to Tamara Broderick, Jenny Huang, Yuzheng Hu, and Jiaqi Ma for making us aware of these prior works via personal communication.

- **Sample-removal strategy:** Following [KATL19], we evaluate both random subsets and more structured sets of training points, selected using heuristics such as clustering by a random feature or by Euclidean distance in feature space.
- **Accuracy metric:** As in [KATL19], we assess accuracy by comparing predicted and actual changes in three scalar quantities when a set $T$ is removed: (1) the total predicted probability for a target class over a subset of test samples, (2) the total test loss on this subset, and (3) the loss on the training set including the removed samples ("self-loss"). The test subset is selected to include a balanced mix of high-loss and randomly chosen test points.
- **Size of removed subset:** We consider values of $|T|$ ranging from $0.1\%$ to $5\%$ of the training set.

We illustrate our main findings in Figure 1. Across every dataset, fraction of sample removals, and accuracy metric, we find that RIF significantly outperforms IF. For more details on our experimental setup, see the supplemental material.

Table 1: Summary of datasets used in our experiments. Each dataset involves a binary classification task which we solve using a regularized logistic regression with mild $L_2$ regularization. We include both datasets used in the [KATL19] benchmark (DogFish and Enron), as well as several new datasets spanning a wide range of domains, including vision, natural language processing, and audio. For more details about these datasets, see the supplementary material.

| Name | $d$ | $n$ | Test Accuracy | Description |
|------|-----|-----|---------------|-------------|
| ESC-50 | 512 | 1600 | 83.0% | ESC-50 dataset embedded using OpenL3; "artificial" vs "natural" classification [Pic15, CWSB19] |
| CatDog | 2048 | 9600 | 80.9% | ResNet-50 embeddings of CIFAR-10 cat and dog classes [Kri09, Tor16] |
| AutoTruck | 2048 | 9600 | 92.7% | ResNet-50 embeddings of CIFAR-10 truck and automobile classes [Kri09, Tor16] |
| DogFish | 2048 | 1800 | 98.3% | Inception v3 embeddings of dog and fish images from ImageNet [SVI+16, RDS+15] |
| Enron | 3294 | 4137 | 96.1% | Bag-of-words embeddings of the standard spam vs ham dataset [KATL19, MAP06] |
| IMDB | 512 | 40000 | 87.7% | BERT embeddings of the IMDB sentiment dataset [MDP+11, DCLT19] |

**Tradeoff: Dimension and Regularization**   As the number of samples $n$ decreases compared to the model dimension $d$, we expect the higher-order effect captured by RIF to be stronger. Figure 2 shows this tradeoff, comparing the IF and RIF accuracy while varying the ratio of $n$ and $d$ by sub-sampling a fixed dataset. A similar tradeoff appears when we add an $L_2$ regularization term of $\frac{1}{2}\lambda\|\theta\|^2$ to the loss for different values of $\lambda > 0$. Increasing $\lambda$ dampens the higher-order effects captured by RIF – in the limit $\lambda \to \infty$ the Hessian does not vary as samples are removed. In Figure 2 we illustrate this tradeoff by varying $\lambda$ for a fixed dataset (DogFish), observing that IF and RIF agree for large $\lambda$ but not for small $\lambda$.

**Detecting Data Poisonings with RIF**   One common use of additive data attributions such as influence functions is to detect potential outliers contaminating a dataset [KL17, BGM20, RH25, KLM+23]. We conduct a simple experiment to demonstrate the advantages of RIF over IF for this task. We take a binary image classification problem (Truck vs Automobile), add an incorrectly-labeled test sample to the training set, and train a logistic regression model on the resulting poisoned dataset. We then compare the accuracy of IF and RIF estimates of the effect that removing the poisoned sample would have on the model's prediction for that test sample. RIF significantly outperforms IF. See Figure 3.

## 3   Theoretical Results

We turn to a theoretical explanation of the effectiveness of RIF to estimate leave-$T$-out effects in high dimensions. Prior work [KATL19] shows that under reasonable assumptions, the NS approximation

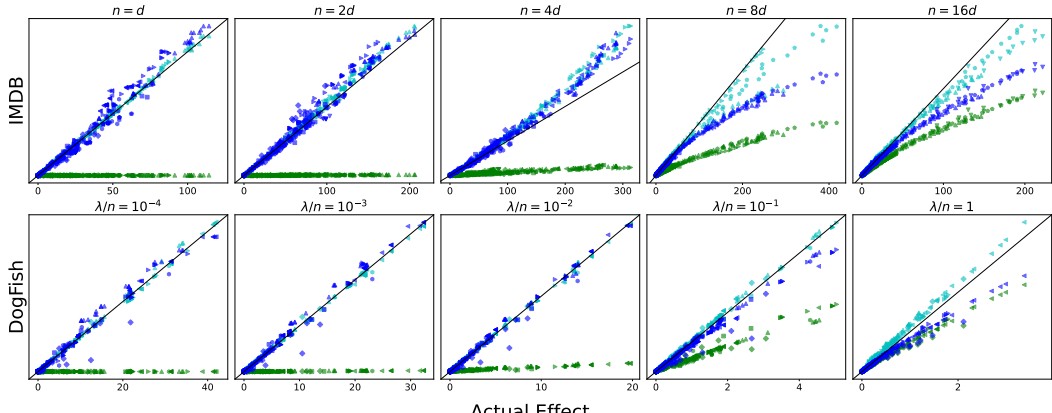

Figure 2: *First row:* accuracy of IF versus RIF compared across differing ratios of $n$ and $d$, for the IMDB dataset, subsampled randomly to obtain datasets of varying sizes. IF and RIF are similar when $n \gg d$, but as $n$ decreases, RIF remains accurate while IF degrades. *Second row:* A similar comparison for the overparameterized DogFish dataset, where we vary the regularization strength $\lambda$. IF becomes accurate only under strong regularization, while RIF remains robust across settings. In all plots, we compare the predicted versus actual values of the self-loss metric. Blue points show the RIF estimate, green points the IF estimate, and cyan points the Newton step. Point shapes indicate different strategies for selecting training samples to remove, as in Figure 1.

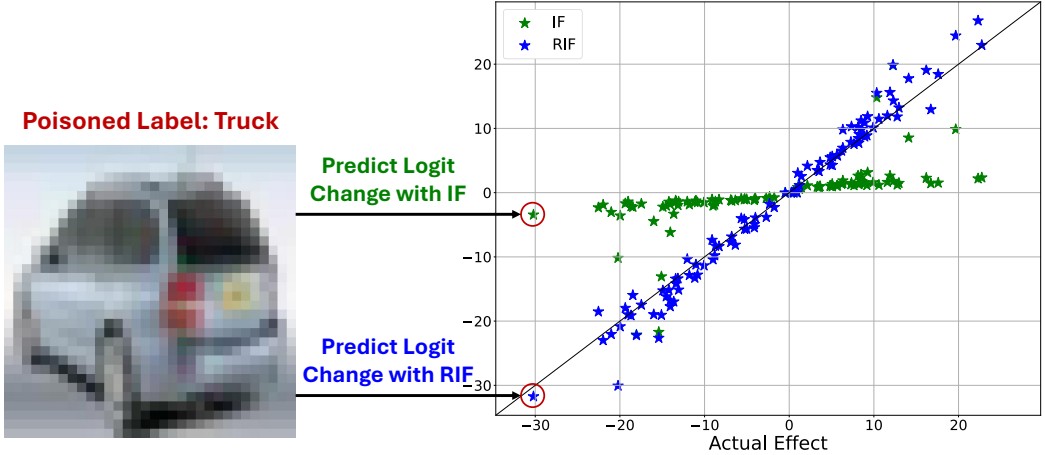

Figure 3: On the right we plot the actual vs predicted effect on a test samples logits from removing a "poisoned" sample from the train set using both IF and RIF. On the left we show the poisoned image corresponding to the leftmost point in the plot – an image of an automobile mislabeled as "Truck". RIF predictions (blue) align much more closely with the actual effects, while IF predictions (green) tend to underestimate these effects.

provides a very accurate approximation of the true leave-$T$-out effect; this is also easily visible in the experiments we reproduced above. Importantly, the NS approximation remains accurate even when the IF estimate is poor. Motivated by this, we focus our analysis on the gap between our RIF estimate and the NS estimate. This leads to a comparatively simple theorem statement, avoiding too many assumptions.

Our setting is as follows. We assume that a model is trained via minimization of a convex empirical risk of the form:

$$\hat{\boldsymbol{\theta}} = \arg\min_{\boldsymbol{\theta} \in \mathbb{R}^d} \sum_{i=1}^{n} \ell_i(\boldsymbol{\theta}) \,.$$

We think of each $\ell_i$ as a per-sample loss from the $i$-th sample in an underlying training set, although we do not actually need to assume such a training set underlies the optimization problem. Let $\mathbf{g}_i := \nabla\ell_i(\hat{\boldsymbol{\theta}})$ and $\mathbf{H}_i := \nabla^2\ell_i(\hat{\boldsymbol{\theta}})$ denote the gradient and Hessian of the $i$th sample at the solution $\hat{\boldsymbol{\theta}}$, and define the total Hessian $\mathbf{H} := \sum_{i=1}^n \mathbf{H}_i$.

We make the following set of assumptions on the loss functions. Most of the assumptions are parameterized quantitatively, and our final theorem bounding the quality of the RIF approximation depends on these parameters. Crucially, these assumptions allow for $n \approx d$ (or even $n \ll d$, if regularization is added), so that our main theorem captures how RIF remains accurate for high-dimensional barely-underparameterized or even overparameterized models. We discuss after our main theorem statement how to interpret these assumptions quantitatively.

**Assumption 1** (Positive Semidefiniteness/Convexity). *We assume that each $\mathbf{H}_i$ is positive semidefinite, or equivalently, that $\ell_i$ is convex.*

The next two assumptions are the key quantitative ones. We offer some discussion now and more after we state our main theorem.

**Assumption 2** (No Single-Sample Gradient or Hessian Too Large). *For all $i \in \{1, \ldots, n\}$, we assume*

$$\left\|\mathbf{H}^{-1/2}\mathbf{g}_i\right\|_2 \le C_\ell \ \text{ and } \ \left\|\mathbf{H}^{-1/2}\mathbf{H}_i\mathbf{H}^{-1/2}\right\|_{\text{op}} \le 1 - \frac{1}{C_R},$$

*for some $C_\ell, C_R > 0$. Here $\|\cdot\|_{\text{op}}$ is the operator norm/maximum singular value.*

The second clause of Assumption 2 can be rewritten as $\mathbf{H}_i \preceq C_R(1 - C_R^{-1})\sum_{j\ne i}\mathbf{H}_j^\top$. This just captures that no single-sample Hessian $\mathbf{H}_i$ is too much larger in any direction than the sum of all the others. This is the key condition allowing for large dimension $d$: even if $n \approx d$, this condition can be satisfied (and indeed will be satisfied for, e.g., random low-rank $\mathbf{H}_i$) without taking $C_R = \omega(1)$.

**Assumption 3** (Cross-Sample Incoherence). *For some $\varepsilon, \delta > 0$, and for all $i \ne j$, $\left\|\mathbf{H}_i^{1/2}\mathbf{H}^{-1}\mathbf{H}_j^{1/2}\right\|_{\text{op}} \le \delta$ and $\left\|\mathbf{H}_i^{1/2}\mathbf{H}^{-1}\mathbf{g}_j\right\|_2 \le \varepsilon$.*

We expect $\varepsilon, \delta$ to be small because in high dimensions gradients and Hessians of distinct samples are likely to point in close-to-orthogonal directions. We carry this intuition out in more detail below.

Ultimately, we use IF/RIF/NS to estimate the change to $f(\hat{\boldsymbol{\theta}})$ for some *evaluation function $f$*. For instance, in our experiments, $f$ is typically test loss or a test prediction. To show that the RIF and NS estimates are close, we require our evaluation function $f$ to have bounded gradients:

**Assumption 4** (Evaluation Gradient Projection Control). *Let $\nabla f(\boldsymbol{\theta})$ denote the gradient of an evaluation function $f\colon \mathbb{R}^d \to \mathbb{R}$. For all $i$, $\left\|\mathbf{H}_i^{1/2}\mathbf{H}^{-1}\nabla f(\hat{\boldsymbol{\theta}})\right\|_2 \le \eta$ for some $\eta > 0$.*

Let $\mathbf{w} \in [0,1]^n$ be a weight change vector. We study the NS and RIF approximations to the optimum of the weighted loss $\sum_{i\le n} w_i\ell_i(\boldsymbol{\theta})$. (So, to capture leave-$T$-out, we set $w_i = 1$ for $i \in T$ and otherwise $w_i = 0$.) We define $\hat{\boldsymbol{\theta}}_{\text{RIF},\mathbf{w}}$ and $\hat{\boldsymbol{\theta}}_{\text{NS},\mathbf{w}}$ analogously to $\hat{\boldsymbol{\theta}}_{\text{RIF},T}, \hat{\boldsymbol{\theta}}_{\text{NS},T}$, respectively. We are now ready to state our main theorem:

**Theorem 3.1** (Accuracy of Rescaled Influence Function). *Under Assumptions 1–4, for any $k \le \frac{1}{2\delta C_R}$,*

$$|\langle \nabla f(\hat{\boldsymbol{\theta}}), \hat{\boldsymbol{\theta}}_{NS,\mathbf{w}} - \hat{\boldsymbol{\theta}}_{RIF,\mathbf{w}}\rangle| \le k^2\eta\,(1 + 2C_R)\,(\varepsilon + C_R C_\ell \delta)$$

The proof of Theorem 3.1 proceeds via a matrix-perturbation analysis which shows that the Hessian inversion in the NS approximation can itself be approximated well without considering the contributions to the inverse from $\nabla^2\ell_i$'s interaction with $\nabla^2\ell_j$ when $i \ne j$. We defer the proof to supplemental material, and focus instead on interpreting Theorem 3.1, to illustrate how it captures the improvement of RIF compared to IF.

**Interpreting Assumptions and Theorem 3.1** Prior works [GSL+19, KATL19] prove similar-in-spirit results to Theorem 3.1, but concerning IF rather than RIF. A direct comparison of Theorem 3.1 to those results in prior work is challenging, as each result is derived under different assumptions. So, to better understand the practical significance of our bounds compared to those in prior work, and see

why they capture the accuracy of RIF for overparameterized models, we analyze their asymptotic behavior in a simplified setting. Since this is for illustration purposes only, we keep the analysis informal.

Consider linear regression with square loss (ordinary least squares), where the data vectors are drawn i.i.d. from a standard Gaussian distribution, $\mathbf{x}_i \sim \mathcal{N}(\mathbf{0}, \mathbf{I})$. And suppose $n \geq (1 + \Omega(1))d$, i.e., $n$ and $d$ are comparable. In this case, we know that:

- Each individual Hessian contribution $\mathbf{H}_i = x_i x_i^\top$ is low rank with $\mathrm{rk}(\mathbf{H}_i) = 1$ and $\|\mathbf{H}_i\|_{\mathrm{op}} = O(d)$,

- The total Hessian is approximately isotropic: $\mathbf{H} \approx n\mathbf{I}$,

- Gradient vectors are bounded in norm: $\|\mathbf{g}_i\|_2 \approx \sqrt{d}$.

We can apply the heuristic that random vectors $u, v \in \mathbb{R}^d$ are likely to have $|\langle u, v \rangle| \approx \|u\|\|v\|/\sqrt{d}$, and so long as $n \geq (1 + \Omega(1))d$, we expect the key variables in Theorem 3.1 to scale as:

- $C_\ell := \max_{i \in [n]} \left\| \mathbf{H}^{-1/2} \mathbf{g}_i \right\|_2 \approx \frac{\sqrt{d}}{\sqrt{n}} = O(1)$,

- $C_R := \max_{i \in [n]} \frac{1}{1 - \left\| \mathbf{H}^{-1/2} \mathbf{H}_i \mathbf{H}^{-1/2} \right\|_{\mathrm{op}}} \approx \frac{n}{n-d} = O(1)$,

- $\delta := \max_{i \neq j} \left\| \mathbf{H}_i^{1/2} \mathbf{H}^{-1} \mathbf{H}_j^{1/2} \right\|_{\mathrm{op}} = \widetilde{O}\left(\frac{\sqrt{d}}{n}\right)$,

- $\varepsilon := \max_{i \neq j} \left\| \mathbf{H}_i^{1/2} \mathbf{H}^{-1} \mathbf{g}_j \right\|_2 = \widetilde{O}\left(\frac{\sqrt{d}}{n}\right)$,

- $\eta := \max_{i \in [n]} \left\| \mathbf{H}_i^{1/2} \mathbf{H}^{-1} \nabla_{\boldsymbol{\theta}} f \right\|_2 = \max_{i \in [n]} \left| \mathbf{x}_i^\intercal \mathbf{H}^{-1} \nabla_{\boldsymbol{\theta}} f \right| = \widetilde{O}\left(\frac{\|\nabla_{\boldsymbol{\theta}} f\|_2}{n}\right)$.

Under these conditions, Theorem 3.1 guarantees that for any set of at most $k \leq k_{\mathrm{threshold}} = \widetilde{\Omega}\left(\frac{n}{\sqrt{d}}\right)$ removed samples, the discrepancy between the RIF and Newton step estimates is bounded by:

$$|\langle \nabla f(\hat{\boldsymbol{\theta}}), \hat{\boldsymbol{\theta}}_{\mathrm{NS},\mathbf{w}} - \hat{\boldsymbol{\theta}}_{\mathrm{RIF},\mathbf{w}} \rangle| \leq k^2 \eta \left(1 + 2C_R\right)\left(\varepsilon + C_R C_\ell \delta\right) = \widetilde{O}\left(\frac{k^2 \sqrt{d}\, \|\nabla_{\boldsymbol{\theta}} f\|_2}{n^2}\right).$$

The scaling rate $n^{-2}$ in the denominator matches what we expect for influence functions, as established in [GSL+19]. But influence function approximations incur *significantly worse* dimension dependence in the numerator, meaning that $n$ must be much larger than $d$ (indeed, quadratic in $d$ or even larger) to obtain nontrivial guarantees. For comparison, in supplemental material, we analyze the bounds proved by [GSL+19, KATL19] for influence functions to the same random-design ordinary-least-squares setting and show that they guarantee influence function accuracy only for much larger $n$ or smaller $d$. For example, the bounds of [GSL+19] are only applicable for $k \leq \widetilde{O}\left(\frac{n}{d^2}\right)$, and yield an error bound that scales as $\widetilde{O}\left(\frac{k^2 d^4 \|\nabla_{\boldsymbol{\theta}} f\|_2}{n^2}\right)$.

Finally, to assess the tightness of our result relative to the RIF magnitude itself, we note that under the same random-design least-squares setup and the same heuristics about inner products of high-dimensional random vectors, the RIF estimate for the removal of the top-$k$ most influential samples scales as

$$\max\left\{ |\langle \nabla f(\hat{\boldsymbol{\theta}}), \hat{\boldsymbol{\theta}}_{\mathrm{RIF},\mathbf{w}} \rangle| \, : \, \|\mathbf{w}\|_1 = k \right\} = \Omega\left(\frac{k\, \|\nabla_{\boldsymbol{\theta}} f\|_2}{n}\right).$$

Hence, the ratio of the RIF estimate ("signal") to the RIF–NS error ("noise") is

$$\mathrm{SNR} := \frac{\max\left\{ |\langle \nabla f(\hat{\boldsymbol{\theta}}), \hat{\boldsymbol{\theta}}_{\mathrm{RIF},\mathbf{w}} \rangle| \, : \, \|\mathbf{w}\|_1 = k \right\}}{\max\left\{ |\langle \nabla f(\hat{\boldsymbol{\theta}}), \hat{\boldsymbol{\theta}}_{\mathrm{NS},\mathbf{w}} - \hat{\boldsymbol{\theta}}_{\mathrm{RIF},\mathbf{w}} \rangle| \, : \, \|\mathbf{w}\|_1 = k \right\}} = \widetilde{\Omega}\left(\frac{n}{k\sqrt{d}}\right).$$

This implies that RIF provides a good relative-error approximation to NS even in high dimensions, provided $k \ll \frac{n}{\sqrt{d}}$.

# 4   Related Work

Influence functions were introduced by Hampel in the context of robust statistics [Ham74], and in the context of estimation of standard errors via the *infinitesimal jackknife* by Jaeckel [Jae72], with a broad ensuing literature in statistics; see e.g., [Law86, GSL+19]. Recent work in econometrics [BGM20] uses influence functions to uncover robustness issues in large empirical studies.

The seminal work [KL17] introduced the modern use of influence functions to study the relationship between training data and model behavior in modern machine learning. Ensuing works [BNL+22, BPF21, GBA+23, FZ20] study influence functions for neural networks, and use them as a tool to study and interpret model behavior. [GJB19, BYF20] propose second and higher-order approximations to leave-one-out and leave-$T$-out effects, but these approximations sacrifice linearity and efficiency. Many applications of influence functions have appeared recently, e.g., machine unlearning [GGHVDM19, SAKS21, SW22], data valuation [JDW+19], robustness quantification [SS19], and fairness [LL22]. To scale influence functions up to very large models and datasets, where Hessian inversion becomes infeasible, several works develop sketching/random projection techniques to approximate influence functions, e.g., [WCZ+16, PGI+23b, SZVS22].

Data attribution – tracing model behavior back to subsets of training data – has become a major industry in machine learning; see the recent survey [HL24] and extensive citations therein, as well as the NeurIPS 2024 workshop [NMI+24] and ICML 2024 tutorial [MIE+24].

Newton-step approximations to the leave-1-out error have been studied since at least 1981 [Pre81]. *Cross-validation* is an especially important application [RM18, WKM20]. Additionally, several recent works consider data models that additively combine estimates of leave-one-out effects to compute a leave-$T$-out effect [KZCC21, IPE+22, PGI+23a, HHZM24, HBN+24]. However, to the best of our knowledge no previous work provides an empirical or theoretical evaluation of the RIF method beyond low-dimensional least-squares regression.

# 5   Discussion and Conclusion

**IFs and Importance-Ordering: Revisiting the Common Wisdom**   Common wisdom regarding IF approximations to leave-$T$-out effects for high-dimensional models holds that the approximations typically *underestimate* the true leave-$T$-out effect, but that there is a strong correlation between the influence-function approximation to the leave-$T$-out effects and the true leave-$T$-out effects, especially measured in terms of the *ordering* of subsets based on their predicted/actual leave-$T$-out effect. The seminal [KATL19] even phrases this as an outstanding open question, writing that their work "opens up the intriguing question of why we observe [correlation and underestimation] across a wide range of empirical settings".

Our work sheds significant light on this question. First of all, it explains why we see such correlation in a great many cases – if most samples have a similar "rescale factor" relating IF and RIF (which we would expect to happen for e.g., random data), this induces a linear relationship between RIF and IF estimates. Since RIF is an excellent approximation to the true leave-$T$-out effect, this explains the correlation between IF and the ground truth, and explains why IF typically underestimates the truth – the rescale factors are always larger than 1.

[KATL19] also note that this IF/ground-truth correlation phenomenon need not be universal, and indeed we observe several experiments where it does not hold. For instance, in the first row of Figure 1, in the Cat vs Dog dataset, we see a dramatically non-linear and even non-monotone relationship between IF and ground truth, since different subset-selection strategies yield very different relationships between IF and ground truth. Even the ordering of subsets by IF-predicted effect is not accurate in this example, but RIF remains accurate.

**Limitations**   Although much more accurate than IFs, RIFs are still imperfect predictors of ground-truth – see e.g., the ESC-50 dataset in Figure 1 or the rightmost variants of the IMBD dataset in Figure 2. We expect high-dimensional logistic regression to be a good "model organism" for high-dimensional machine learning, so our experiments are limited to that setting. RIF also still requires inverting the Hessian; as discussed in related work for very large-scale models this can be computationally infeasible, and approximate techniques are required. While we show that RIFs are

preferable to IFs for detecting certain simple data-poisoning attacks, we do not expect that RIFs are a secure general defense against data poisoning.

**Conclusion**   We show that RIFs are an appealing drop-in replacement for IFs, with little computational overhead in generalized linear models (or whenever individual training samples contribute low-rank terms to the Hessian), but dramatically improved accuracy. Both experiments and theory support this conclusion. Furthermore, the fact that RIFs and IFs differ by a per-sample scaling factor helps to resolve an open question from prior work, showing that the correlation between IF and ground truth leave-$T$-out occurs when the per-sample scalings all (approximately) agree.

## Acknowledgments and Disclosure of Funding

We would like to thank Jenny Y. Huang, David R. Burt, Yunyi Shen, Tin D. Nguyen, Vishwak Srinivasan, Tamara Broderick, Yuzheng Hu and Jiaqi Ma for helpful conversations and correspondences. This work was supported by NSF Award No. 2238080 and CSAIL Alliances.

## Compute Resources

All experiments were conducted on a server equipped with 64GB RAM, 2 IBM POWER9 CPU cores, and 4 NVIDIA Tesla V100 SXM2 GPUs (each with 32GB memory).

Table 2 details the computational cost of training the base models and computing their IF and RIF data attribution. Another major computational overhead was in retraining the model to obtain ground-truth values for the retrain effect. Despite this, compute resources were not a bottleneck for our work. The total wall-clock time for all experiments reported in the paper was under 100 hours.

Table 2: Comparison of runtime components across datasets. The **rescaling** step consistently added negligible overhead across all experiments.

| Dataset | Training | Hessian | Inversion | Influence | Rescaling |
|---|---|---|---|---|---|
| ESC50 | 1.8 s | 0.056 s | 0.0005 s | 0.051 s | 0.0033 s (0.2%) |
| CatDog | 76 s | 4.9 s | 0.010 s | 4.8 s | 0.087 s (0.1%) |
| AutoTruck | 48 s | 4.9 s | 0.0094 s | 4.8 s | 0.087 s (0.2%) |
| DogFish | 0.43 s | 0.92 s | 0.0095 s | 0.89 s | 0.015 s (0.7%) |
| Enron | 6.7 s | 15 s | 0.065 s | 15 s | 0.095 s (0.3%) |
| IMDB (n=16d) | 20 s | 0.92 s | 0.0012 s | 0.87 s | 0.044 s (0.2%) |

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

# A    Proof of Theorem 3.1

Recall our main theoretical result from Section 3:

**Theorem A.1** (Theorem 3.1 (restated))**.** *Under Assumptions 1–4, for any $k \leq \frac{1}{2\delta C_R}$,*

$$|\langle \nabla f(\hat{\boldsymbol{\theta}}), \hat{\boldsymbol{\theta}}_{NS,\mathbf{w}} - \hat{\boldsymbol{\theta}}_{RIF,\mathbf{w}} \rangle| \leq k^2 \eta \left(1 + 2C_R\right) \left(\varepsilon + C_R C_\ell \delta\right)$$

Before delving into the proof of Theorem 3.1, we introduce a useful technical lemma:

**Lemma A.2.** *Let $\mathbf{A}_1, \ldots, \mathbf{A}_k \in \mathbb{R}^{d \times d}$ and let $\mathbf{H} \in \mathbb{R}^{d \times d}$ be positive semidefinite. Suppose:*

- $\left\| \mathbf{H}^{-1/2} \mathbf{A}_i \mathbf{H}^{-1/2} \right\|_{\mathrm{op}} \leq \sigma$ *for all $i$,*

- $\left\| \sqrt{\mathbf{A}_i} \mathbf{H}^{-1} \sqrt{\mathbf{A}_j} \right\|_{\mathrm{op}} \leq \delta_{ij}$ *for all $i \neq j$.*

*Then,*

$$\left\| \sum_{i=1}^{k} \mathbf{H}^{-1/2} \mathbf{A}_i \mathbf{H}^{-1/2} \right\|_{\mathrm{op}} \leq \sigma + \sqrt{\sum_{i \neq j} \delta_{ij}^2}.$$

*Proof of Theorem 3.1.* We begin by analyzing the difference between the Newton step and the rescaled influence function (RIF) approximation.

Recall that the Newton step is defined as:

$$\text{Newton Step} = (\nabla f)^\top \left( \mathbf{H} - \sum_{j=1}^{n} w_j \mathbf{H}_j \right)^{-1} \sum_{i=1}^{n} w_i \mathbf{g}_i,$$

where each $\mathbf{g}_i \in \mathbb{R}^d$ is the $i$th gradient component, and $\mathbf{H}_i$ is the $i$th contribution to the Hessian. Define the weighted Hessian:

$$\mathbf{H}_{\mathbf{w}} := \mathbf{H} - \sum_{j=1}^{n} w_j \mathbf{H}_j.$$

For each $i \in \{1, \ldots, n\}$, define $\mathbf{w}^{(i)} := w \cdot \mathbb{1}_{\{i\}}$ to isolate the $i$-th coordinate. The RIF estimator is given by:

$$\text{RIF}_i = \sum_{i=1}^{n} (\nabla f)^\top \mathbf{H}_{\mathbf{w}^{(i)}}^{-1} w_i \mathbf{g}_i \,.$$

Our goal is to bound the difference between the Newton step and RIF estimators and we do this by bounding the contribution of each individual sample. That is, for each $i \in [n]$, we will try to bound

$$(\nabla f)^\top \left( \mathbf{H}_{\mathbf{w}}^{-1} - \mathbf{H}_{\mathbf{w}^{(i)}}^{-1} \right) \mathbf{g}_i \,.$$

To do so, we begin by expressing each matrix in terms of $\mathbf{H}$ and its perturbations. Observe:

$$\mathbf{H_w} = \mathbf{H}^{1/2} \left( I - \mathbf{G_w} \right) \mathbf{H}^{1/2}, \quad \text{where } \mathbf{G_w} := \sum_j \mathbf{H}^{-1/2} w_j \mathbf{H}_j \mathbf{H}^{-1/2}.$$

Moreover, we define $\mathbf{R} := \left( I - \mathbf{G}_{\mathbf{w}^{(i)}} \right)^{-1}$, where $\mathbf{G}_{\mathbf{w}^{(i)}} = \mathbf{H}^{-1/2} w_i \mathbf{H}_i \mathbf{H}^{-1/2}$. We have

$$\mathbf{H}_{\mathbf{w}^{(i)}} = \mathbf{H}^{1/2} \left( I - \mathbf{G}_{\mathbf{w}^{(i)}} \right) \mathbf{H}^{1/2}.$$

Using the matrix identity:

$$(A - B)^{-1} = A^{-1} + (A - B)^{-1} B A^{-1},$$

with $A = \mathbf{H}_{\mathbf{w}^{(i)}}$, $B = \mathbf{H}_{\mathbf{w}^{(i)}} - \mathbf{H_w}$, we obtain:

$$\mathbf{H_w}^{-1} = \mathbf{H}_{\mathbf{w}^{(i)}}^{-1} + \mathbf{H_w}^{-1} \left( \mathbf{H}_{\mathbf{w}^{(i)}} - \mathbf{H_w} \right) \mathbf{H}_{\mathbf{w}^{(i)}}^{-1}.$$

We now expand the correction term on the right-hand side further by applying the same identity again, this time expanding $\mathbf{H_w} = \mathbf{H} - (\mathbf{H} - \mathbf{H_w})$,

$$\mathbf{H_w}^{-1} \left( \mathbf{H}_{\mathbf{w}^{(i)}} - \mathbf{H_w} \right) \mathbf{H}_{\mathbf{w}^{(i)}}^{-1} = \mathbf{H}^{-1} \left( \mathbf{H}_{\mathbf{w}^{(i)}} - \mathbf{H_w} \right) \mathbf{H}_{\mathbf{w}^{(i)}}^{-1} + \mathbf{H}^{-1} \left( \mathbf{H} - \mathbf{H_w} \right) \mathbf{H_w}^{-1} \left( \mathbf{H}_{\mathbf{w}^{(i)}} - \mathbf{H_w} \right) \mathbf{H}_{\mathbf{w}^{(i)}}^{-1},$$

where the second term reflects higher-order correction contributions due to recursive matrix inversion.

To bound the full error

$$(\nabla f)^\top \left( \mathbf{H_w}^{-1} - \mathbf{H}_{\mathbf{w}^{(i)}}^{-1} \right) \mathbf{g}_i = (\nabla f)^\top \mathbf{H}^{-1} \left( \mathbf{H}_{\mathbf{w}^{(i)}} - \mathbf{H_w} \right) \mathbf{H}_{\mathbf{w}^{(i)}}^{-1} \mathbf{g}_i +$$
$$+ (\nabla f)^\top \mathbf{H}^{-1} \left( \mathbf{H} - \mathbf{H_w} \right) \mathbf{H_w}^{-1} \left( \mathbf{H}_{\mathbf{w}^{(i)}} - \mathbf{H_w} \right) \mathbf{H}_{\mathbf{w}^{(i)}}^{-1} \mathbf{g}_i .$$

It suffices to control the size of each of these terms separately. In other words, we will proceed to bound:

1. The first order correction $(\nabla f)^\top \mathbf{H}^{-1} \left( \mathbf{H}_{\mathbf{w}^{(i)}} - \mathbf{H_w} \right) \mathbf{H}_{\mathbf{w}^{(i)}}^{-1} \mathbf{g}_i$,

2. The higher order terms $(\nabla f)^\top \mathbf{H}^{-1} \left( \mathbf{H} - \mathbf{H_w} \right) \mathbf{H_w}^{-1} \left( \mathbf{H}_{\mathbf{w}^{(i)}} - \mathbf{H_w} \right) \mathbf{H}_{\mathbf{w}^{(i)}}^{-1} \mathbf{g}_i$.

**Bounding the First Order Correction**

To bound the first order correction, we use the same formula above to split $\mathbf{H}_{\mathbf{w}^{(i)}}^{-1}$ into a leading term and higher order terms. The goal of this separation is to show that this update to the Hessian does not rotate too much of the weight of $\mathbf{g}_i$ onto the eigenspace of $\mathbf{H}_j$ for any $j \neq i$

We have $\mathbf{H}_{\mathbf{w}^{(i)}}^{-1} = \mathbf{H}^{-1} + \mathbf{H}^{-1} w_i \mathbf{H}_i \mathbf{H}_{\mathbf{w}^{(i)}}^{-1}$.

Therefore, for any $j \neq i$,

$$\left\| \mathbf{H}_j^{1/2} \mathbf{H}_{\mathbf{w}^{(i)}}^{-1} \mathbf{g}_i \right\|_2 \leq \underbrace{\left\| \mathbf{H}_j^{1/2} \mathbf{H}^{-1} \mathbf{g}_i \right\|_2}_{\leq \varepsilon} + \underbrace{\left\| w_i \mathbf{H}_j^{1/2} \mathbf{H}^{-1} \mathbf{H}_i \mathbf{H}^{-1/2} \mathbf{R} \mathbf{H}^{-1/2} \mathbf{g}_i \right\|_2}_{\leq |w_i| \delta C_R C_\ell \leq \delta C_R C_\ell} \leq \varepsilon + \delta C_R C_\ell$$

Therefore, this first order correction is at most

$$\sum_{j \neq i} w_j \nabla f^\top \mathbf{H}^{-1} \mathbf{H}_j \mathbf{H}_{\mathbf{w}^{(i)}}^{-1} \mathbf{g}_i \leq \sum_{j \neq i} w_j \underbrace{\left\| \nabla f^\top \mathbf{H}^{-1} \mathbf{H}_j^{1/2} \right\|_2}_{\leq \eta} \underbrace{\left\| \mathbf{H}_j^{1/2} \mathbf{H}_{\mathbf{w}^{(i)}}^{-1} \mathbf{g}_i \right\|_2}_{\leq \varepsilon + C_R C_\ell \delta} \leq k \eta \left( \varepsilon + C_R C_\ell \delta \right)$$

**Bounding the Higher Order Corrections**

We next bound the second (higher-order) term using the Cauchy-Schwarz inequality.

$$\left| (\nabla f)^\top \mathbf{H}^{-1} \left( \mathbf{H} - \mathbf{H_w} \right) \mathbf{H_w}^{-1} \left( \mathbf{H}_{\mathbf{w}^{(i)}} - \mathbf{H_w} \right) \mathbf{H}_{\mathbf{w}^{(i)}}^{-1} \mathbf{g}_i \right| \leq$$
$$\leq \left\| (\nabla f)^\top \mathbf{H}^{-1} \left( \mathbf{H} - \mathbf{H_w} \right) \mathbf{H}^{-1/2} \right\|_2 \times \left\| \left( \mathbf{I} - \mathbf{G_w} \right)^{-1} \right\|_{\text{op}} \times \left\| \mathbf{H}^{-1/2} \left( \mathbf{H}_{\mathbf{w}^{(i)}} - \mathbf{H_w} \right) \mathbf{H}_{\mathbf{w}^{(i)}}^{-1} \mathbf{g}_i \right\|_2$$

We will bound each of these terms independently.

The right-most multiplicand is bounded using the analysis of the first order term

$$\left\| \mathbf{H}^{-1/2} \left( \mathbf{H}_{\mathbf{w}^{(i)}} - \mathbf{H}_{\mathbf{w}} \right) \mathbf{H}_{\mathbf{w}^{(i)}}^{-1} \mathbf{g}_i \right\|_2 \leq \sum_{j \neq i} \left\| \mathbf{H}^{-1/2} w_j \mathbf{H}_j^{1/2} \mathbf{H}_j^{1/2} \mathbf{H}_{\mathbf{w}^{(i)}}^{-1} \mathbf{g}_i \right\|_2 \leq$$

$$\leq \sum_{j \neq i} \left\| w_j \mathbf{H}_j^{1/2} \mathbf{H}_{\mathbf{w}^{(i)}}^{-1} \mathbf{g}_i \right\|_2 \leq k \left( \varepsilon + C_R C_\ell \delta \right)$$

From the triangle inequality,

$$\left\| \sum_{j \neq i} \nabla f^\top \mathbf{H}^{-1} \mathbf{H}_j \mathbf{H}^{-1/2} \right\| \leq \sum_{j \neq i} |w_j| \cdot \left\| \nabla f^\top \mathbf{H}^{-1} \mathbf{H}_j^{1/2} \right\| \cdot \left\| \mathbf{H}_j^{1/2} \mathbf{H}^{-1/2} \right\|_{\text{op}}.$$

Using the assumption $\left\| \mathbf{H}^{-1/2} \mathbf{H}_j \mathbf{H}^{-1/2} \right\|_{\text{op}} \leq 1$, it follows that

$$\left\| \mathbf{H}_j^{1/2} \mathbf{H}^{-1/2} \right\|_{\text{op}} \leq 1,$$

and from Assumption 5, we also have

$$\left\| \nabla f^\top \mathbf{H}^{-1} \mathbf{H}_j^{1/2} \right\| \leq \eta.$$

Therefore,

$$\left\| \sum_{j \neq i} \nabla f^\top \mathbf{H}^{-1} \mathbf{H}_j \mathbf{H}^{-1/2} \right\| \leq \eta \sum_{j \neq i} |w_j| \leq \eta \left\| w \right\|_1 = \eta k.$$

Next, define $\mathbf{A}_j = w_j \mathbf{H}^{-1/2} \mathbf{H}_j \mathbf{H}^{-1/2}$. Then for all $j$,

$$\left\| \mathbf{H}^{-1/2} \mathbf{A}_j \mathbf{H}^{-1/2} \right\|_{\text{op}} = |w_j| \cdot \left\| \mathbf{H}^{-1/2} \mathbf{H}_j \mathbf{H}^{-1/2} \right\|_{\text{op}} \leq 1 - \frac{1}{C_R},$$

since $\left\| w \right\|_\infty \leq 1$ and by Assumption 2 $\left\| \mathbf{H}^{-1/2} \mathbf{H}_j \mathbf{H}^{-1/2} \right\|_{\text{op}} \leq 1 - \frac{1}{C_R}$.

Moreover, for all $i \neq j$, we have

$$\left\| \sqrt{\mathbf{A}_i} \mathbf{H}^{-1} \sqrt{\mathbf{A}_j} \right\|_{\text{op}} \leq \sqrt{|w_i|} \cdot \sqrt{|w_j|} \cdot \delta_{ij}.$$

So,

$$\sum_{i \neq j} \left\| \sqrt{\mathbf{A}_i} \mathbf{H}^{-1} \sqrt{\mathbf{A}_j} \right\|_{\text{op}}^2 \leq \sum_{i \neq j} |w_i| |w_j| \delta_{ij}^2 \leq \left( \left\| w \right\|_1 \right)^2 \cdot \delta^2 = k^2 \delta^2.$$

Applying Lemma A.2 to the collection $\{ \mathbf{A}_j \}$, we conclude that

$$\left\| \mathbf{G}_{\mathbf{w}} \right\|_{\text{op}} \leq 1 - \frac{1}{C_R} + k \delta.$$

For any $k < \frac{1}{2\delta C_R}$, it follows that $I - \mathbf{G}_{\mathbf{w}}$ is PSD and $\left\| \mathbf{G}_{\mathbf{w}} \right\|_{\text{op}} < 1$, so we have

$$\left\| (I - \mathbf{G}_{\mathbf{w}})^{-1} \right\|_{\text{op}} \leq \frac{1}{\frac{1}{C_R} - k\delta} \leq 2 C_R.$$

**Summary:**

So far, we have show that for all $i \in [n]$,

$$\left| (\nabla f)^\top \left( \mathbf{H}_{\mathbf{w}}^{-1} - \mathbf{H}_{\mathbf{w}^{(i)}}^{-1} \right) \mathbf{g}_i \right| \leq \eta k \left( \varepsilon + C_R C_\ell \delta \right) + \eta k \times 2 C_R \times \left( \varepsilon + C_R C_\ell \delta \right).$$

Therefore,

$$\left| \text{Newton Step} - \text{RIF} \right| = \left| \sum_{i=1}^n w_i (\nabla f)^\top \left( \mathbf{H}_{\mathbf{w}}^{-1} - \mathbf{H}_{\mathbf{w}^{(i)}}^{-1} \right) \mathbf{g}_i \right| \leq k^2 \eta \left( 1 + 2 C_R \right) \left( \varepsilon + C_R C_\ell \delta \right)$$

$\square$

*Proof of Lemma A.2.* We define the linear operator $C : \mathbb{R}^{k \times k \times d \times d} \to \mathbb{R}^{d \times d}$ to be

$$C(\mathbf{M}) := \sum_{i,j} \mathbf{H}^{-1/2} \sqrt{\mathbf{A}_i}\, \mathbf{M}_{ij}\, \sqrt{\mathbf{A}_j} \mathbf{H}^{-1/2},$$

where $\mathbf{M} \in \mathbb{R}^{k \times k \times d \times d}$ is a rank-4 tensor with $\mathbf{M}_{ij} \in \mathbb{R}^{d \times d}$.

For tensors $\mathbf{M}$, $\mathbf{N}$, define their contraction:

$$C(\mathbf{M})C(\mathbf{N}) = C(\mathbf{L}), \quad \text{where } \mathbf{L}_{ij} = \sum_{q,r} \mathbf{M}_{iq} \cdot \sqrt{\mathbf{A}_q} \mathbf{H}^{-1} \sqrt{\mathbf{A}_r} \cdot \mathbf{N}_{rj}.$$

Define $\Sigma : \mathbb{R}^{k \times k \times d \times d} \to \mathbb{R}^{k \times k}$ as $\Sigma(\mathbf{M})_{ij} := \|\mathbf{M}_{ij}\|_{\mathrm{op}}$, and define $\mathbf{\Delta} \in \mathbb{R}^{k \times k}$ with entries $\mathbf{\Delta}_{ij} = \left\|\sqrt{\mathbf{A}_i} \mathbf{H}^{-1} \sqrt{\mathbf{A}_j}\right\|_{\mathrm{op}}$. Then by the triangle inequality and submultiplicativity of the operator norm, we have the point-wise inequality

$$\Sigma(\mathbf{L}) \le \Sigma(\mathbf{M}) \cdot \mathbf{\Delta} \cdot \Sigma(\mathbf{N}).$$

Applying this iteratively for a sequence $\mathbf{M}_1, \ldots, \mathbf{M}_\ell$, we obtain:

$$\Sigma(\mathbf{N}) \le \Sigma(\mathbf{M}_1) \cdot \mathbf{\Delta} \cdot \Sigma(\mathbf{M}_2) \cdot \mathbf{\Delta} \cdots \mathbf{\Delta} \cdot \Sigma(\mathbf{M}_\ell).$$

Now consider the identity tensor $\mathbf{M}$ with $\mathbf{M}_{ii} = I_d$ and $\mathbf{M}_{ij} = 0$ for $i \ne j$. Then:

$$C(\mathbf{M}) = \sum_i \mathbf{H}^{-1/2} \sqrt{\mathbf{A}_i} I_d \sqrt{\mathbf{A}_i} \mathbf{H}^{-1/2} = \sum_i \mathbf{H}^{-1/2} \mathbf{A}_i \mathbf{H}^{-1/2}.$$

Let $C := C(\mathbf{M})$. Then:

$$C^\ell = C(\mathbf{M})^\ell = C(\mathbf{N}), \quad \text{with } \Sigma(\mathbf{N}) \le \mathbf{\Delta}^\ell.$$

By triangle inequality and bounding each tensor entry:

$$\left\|C^\ell\right\|_{\mathrm{op}} \le k^2 d^2 \cdot \max_i \left\|\mathbf{H}^{-1/2} \mathbf{A}_i^{1/2}\right\|_{\mathrm{op}}^2 \cdot \left\|\mathbf{\Delta}^\ell\right\|_{\mathrm{op}} \le k^2 d^2 \sigma \cdot \|\mathbf{\Delta}\|_{\mathrm{op}}^\ell.$$

Taking $\ell$-th roots:

$$\|C\|_{\mathrm{op}} \le (k^2 d^2 \sigma)^{1/\ell} \cdot \|\mathbf{\Delta}\|_{\mathrm{op}}.$$

Letting $\ell \to \infty$, the prefactor tends to 1, giving:

$$\left\|\sum_i \mathbf{H}^{-1/2} \mathbf{A}_i \mathbf{H}^{-1/2}\right\|_{\mathrm{op}} \le \|\mathbf{\Delta}\|_{\mathrm{op}}.$$

Now bound $\|\mathbf{\Delta}\|_{\mathrm{op}}$. Each diagonal entry $\mathbf{\Delta}_{ii} = \left\|\sqrt{\mathbf{A}_i} \mathbf{H}^{-1} \sqrt{\mathbf{A}_i}\right\|_{\mathrm{op}} = \left\|\mathbf{H}^{-1/2} \mathbf{A}_i \mathbf{H}^{-1/2}\right\|_{\mathrm{op}} \le \sigma$. Thus,

$$\mathbf{\Delta} = D + R, \quad \text{with } D = \mathrm{diag}\left(\left\|\mathbf{H}^{-1/2} \mathbf{A}_1 \mathbf{H}^{-1/2}\right\|_{\mathrm{op}}, \ldots\right), \quad \|D\|_{\mathrm{op}} \le \sigma.$$

Then:

$$\|\mathbf{\Delta}\|_{\mathrm{op}} \le \|D\|_{\mathrm{op}} + \|R\|_{\mathrm{op}} \le \sigma + \|R\|_{\mathrm{F}},$$

where $R$ is the off-diagonal part of $\mathbf{\Delta}$ and $\|R\|_{\mathrm{F}}^2 = \sum_{i \ne j} \delta_{ij}^2$.

Hence:

$$\left\|\sum_{i=1}^k \mathbf{H}^{-1/2} \mathbf{A}_i \mathbf{H}^{-1/2}\right\|_{\mathrm{op}} \le \sigma + \sqrt{\sum_{i \ne j} \delta_{ij}^2}.$$

$\square$

# B Asymptotic Analyses of the Bounds of [KATL19] and [GSL$^+$19]

## B.1 Analysis of [KATL19]

Koh et al. [KATL19] present two main theoretical results. The first bounds the difference between a single Newton step and a full retrain, and the second bounds the difference between the Newton step and the influence function estimate. We focus on the latter, since that is more directly comparable to the guarantees of Theorem 3.1. To facilitate a direct comparison, we restate their Proposition 2 with all assumptions made explicit below.

**Proposition B.1** (Proposition 2 of [KATL19], rephrased)**.** *Assume the evaluation function $f(\theta)$ is $C_f$-Lipschitz, the Hessian $\nabla_\theta^2 \ell(x, y, \theta)$ is $C_H$-Lipschitz, and the third derivative of $f(\theta)$ exists and is bounded in norm by $C_{f,3}$. Let $\sigma_{\min}$ and $\sigma_{\max}$ be the smallest and largest eigenvalues of $H_1$, respectively, and define*

$$C_\ell \triangleq \max_{1 \le i \le n} \left\| \nabla_\theta \ell(x_i, y_i; \hat{\theta}(1)) \right\|_2 .$$

*Then the Newton-influence error $\mathrm{Err}_{\mathrm{Nt\text{-}inf}}(w)$ is*

$$\mathrm{Err}_{\mathrm{Nt\text{-}inf}}(w) = \nabla_\theta f(\hat{\theta}(1))^\top \mathbf{H}_{\lambda,1}^{-1/2} \mathbf{D}(w) \mathbf{H}_{\lambda,1}^{-1/2} \mathbf{g}(\mathbf{w}) + \underbrace{\frac{1}{2} \Delta\theta_{\mathrm{Nt}}(w)^\top \nabla_\theta^2 f(\hat{\theta}(1)) \Delta\theta_{\mathrm{Nt}}(w) + \mathrm{Err}_{f,3}(w)}_{\textit{Error from the curvature of } f(\cdot)},$$

*where*

$$\mathbf{D}(w) \stackrel{\mathrm{def}}{=} \left( I - H_{\lambda,1}^{-1/2} H_1(w) H_{\lambda,1}^{-1/2} \right)^{-1} - I, \quad \text{and} \quad H_1(w) \stackrel{\mathrm{def}}{=} \sum_{i=1}^{n} w_i \nabla_\theta^2 \ell(x_i, y_i; \hat{\theta}(1)).$$

*The matrix $\mathbf{D}(w)$ has eigenvalues between 0 and $\sigma_{\max}/\lambda$. The residual term $\mathrm{Err}_{f,3}(w)$ captures the error due to third-order derivatives and is bounded by*

$$|\mathrm{Err}_{f,3}(w)| \le \|w\|_1^3 C_{f,3} C_\ell^3 / \left( 6(\sigma_{\min} + \lambda)^3 \right).$$

To compare this guarantee with Theorem 3.1, which bounds the inner product between the data attribution error and $\nabla f$, we focus on the first term in the bound from Proposition B.1. This term quantifies the error in estimating the linear evaluation function $f$ using influence functions.

Recall that in the simple linear regression setting we define for our simplified asymptotic analysis, we have $\mathbf{H} \approx n\mathbf{I}$, and this is also the case with $\mathbf{H}_{\lambda,1}$. Using the bound $\mathbf{D}(w) \preceq \frac{\sigma_{\max}}{\lambda} \mathbf{I}$ from Proposition B.1, the Cauchy–Schwarz inequality gives:

$$\left| \nabla_\theta f(\hat{\theta}(1))^\top \mathbf{H}_{\lambda,1}^{-1/2} \mathbf{D}(w) \mathbf{H}_{\lambda,1}^{-1/2} \mathbf{g}(\mathbf{w}) \right| \lesssim \frac{\sigma_{\max}}{n\lambda} \left\| \nabla_\theta f(\hat{\theta}(1)) \right\|_2 \|\mathbf{g}(\mathbf{w})\|_2 .$$

The scaling of $\sigma_{\max}/\lambda$ depends on the regime. Under strong regularization (e.g., bottom-right of Figure 2), it may be $O(1)$. However, as Koh et al. observe, this rarely happens in practice, suggesting that it would be more reasonable to assume that $\sigma_{\max}/\lambda = \omega(1)$.

Let $\mathbf{g}$ denote the per-sample gradient, so that $\mathbf{g}(\mathbf{w}) = \sum_i w_i \mathbf{g}_i$ represents the total gradient over removed samples. Following Koh et al.'s approach in Proposition 1, we apply the triangle inequality to bound

$$\|\mathbf{g}(\mathbf{w})\|_2 \le \|\mathbf{w}\|_1 \max_{i \in [n]} \{\|\mathbf{g}_i\|_2\} = \Theta(k\sqrt{d}).$$

Altogether, the Koh et al. bound on the difference between the IF and the NS estimations for the 1st order change in $f$ comes out to

$$\frac{\sigma_{\max}}{n\lambda} \left\| \nabla_\theta f(\hat{\theta}(1)) \right\|_2 \|\mathbf{g}(\mathbf{w})\|_2 = \omega \left( \frac{k\sqrt{d}}{n} \right) \times \left\| \nabla_\theta f(\hat{\theta}(1)) \right\|_2$$

To get a sense for the scaling of this bound, as with the bound of Theorem 3.1, we compare it to the actual IF estimate to obtain an estimate of signal-to-noise-ratio between IF and its distance from NS

$$\mathrm{SNR} = \frac{\max_{\|\mathbf{w}\|_1 \le k} \left\{ \left| \langle \nabla_\theta f, \boldsymbol{\theta}_\mathbf{w}^{\mathrm{IF}} - \boldsymbol{\theta}_\mathbf{w} \rangle \right| \right\}}{\mathrm{Err}_{\mathrm{Nt\text{-}inf}}(w)} = \Theta \left( \frac{\lambda}{\sigma_{\max}} \right) = o(1).$$

Therefore, the guarantee of Koh et al. do not rule out the possibility of the difference between the NS estimate and the IF estimate completely dominating the removal effects even in simple scenarios (regardless of how $k, d$ may scale with $n$).

## B.2 Analysis of [GSL+19]

### B.2.1 Assumptions and Statement

We now summarize the theoretical guarantees provided by Giordano et al., which underlie their infinitesimal jackknife approximation for estimating the effect of data perturbations.

**Assumption 5** (Smoothness; Assumption 1 of [GSL+19]). *For all $\theta \in \Omega_\theta$, each $g_n(\theta)$ is continuously differentiable in $\theta$.*

**Assumption 6** (Non-degeneracy; Assumption 2 of [GSL+19]). *For all $\theta \in \Omega_\theta$, the Hessian $H(\theta, \mathbf{1}_w)$ is non-singular, with*

$$\sup_{\theta \in \Omega_\theta} \left\| H(\theta, \mathbf{1}_w)^{-1} \right\|_{op} \le C_{op} < \infty.$$

**Assumption 7** (Bounded averages; Assumption 3 of [GSL+19]). *There exist finite constants $C_g$ and $C_h$ such that*

$$\sup_{\theta \in \Omega_\theta} \frac{1}{\sqrt{N}} \left\| g(\theta) \right\|_2 \le C_g \quad and \quad \sup_{\theta \in \Omega_\theta} \frac{1}{\sqrt{N}} \left\| h(\theta) \right\|_2 \le C_h.$$

**Assumption 8** (Local smoothness; Assumption 4 of [GSL+19]). *There exists $\Delta_\theta > 0$ and a finite constant $L_h$ such that for all $\theta$ with $\|\theta - \hat{\theta}_1\|_2 \le \Delta_\theta$,*

$$\frac{1}{\sqrt{N}} \left\| h(\theta) - h(\hat{\theta}_1) \right\|_2 \le L_h \left\| \theta - \hat{\theta}_1 \right\|_2.$$

**Assumption 9** (Bounded weight averages; Assumption 5 of [GSL+19]). *The weighted norm $\frac{1}{\sqrt{N}}\|w\|_2$ is uniformly bounded for $w \in W$ by a constant $C_w < \infty$.*

**Condition 1** (Set complexity; Condition 1 of [GSL+19]). *There exists a $\delta \ge 0$ and a corresponding subset $W_\delta \subseteq W$ such that:*

$$\max_{w \in W_\delta} \sup_{\theta \in \Omega_\theta} \left\| \frac{1}{N} \sum_{n=1}^{N} (w_n - 1) g_n(\theta) \right\|_1 \le \delta, \quad and \quad \max_{w \in W_\delta} \sup_{\theta \in \Omega_\theta} \left\| \frac{1}{N} \sum_{n=1}^{N} (w_n - 1) h_n(\theta) \right\|_1 \le \delta.$$

**Definition 1** (Constants from Assumptions). *Define*

$$C_{IJ} := 1 + D C_w L_h C_{op}, \quad and \quad \Delta_\delta := \min \left\{ \Delta_\theta C_{op}^{-1}, \tfrac{1}{2} C_{IJ}^{-1} C_{op}^{-1} \right\}.$$

**Theorem B.2** (Error bound for the approximation; Theorem 1 of [GSL+19]). *Under Assumptions 5–9, if $\delta \le \Delta_\delta$, then*

$$\max_{w \in W_\delta} \left\| \hat{\theta}_{IJ}(w) - \hat{\theta}(w) \right\|_2 \le 2 C_{op}^2 C_{IJ} \delta^2.$$

### B.2.2 Analysis

We now analyze the guarantees provided by Giordano et al. [GSL+19] in the context of our linear regression setting.

In our setup with squared loss and a linear model, the first- and second-order statistics become:

$$g_i(\theta) = x_i(y_i - \langle x_i, \theta \rangle), \qquad h_i(\theta) = x_i x_i^\top.$$

Note that $h_i(\theta)$ does not depend on $\theta$, and thus the local smoothness constant $L_h$ (Assumption 8) is zero. Further, the Hessian takes the form

$$H(\theta, w) = \frac{1}{n} \sum_{i=1}^{n} w_i x_i x_i^\top,$$

so assuming the data is appropriately scaled, we expect the spectrum of its Hessian to be somewhat clustered and hence $C_{op} = O(1)$ (Assumption 6).

Assumption 7 requires bounds on $\|g(\theta)\|_2$ and $\|h(\theta)\|_2$. In general, linear regression does not admit uniform convergence over $\theta$ due to unbounded gradients as $\theta \to \infty$, but if we fix $\|\theta\|$ to a moderate scale by limiting the scope of $\Omega_\theta$, we can reasonably assume that $\|g_i(\theta)\|_2 \approx \sigma\sqrt{d}$, giving $C_g \approx \sigma\sqrt{d} = O(\sqrt{d})$ and $C_h \approx d$.

We now turn to Condition 1, which controls how large the weighted deviations can be. In particular, we focus on the second half of this condition, which requires that

$$\max_{w \in W_\delta} \sup_{\theta \in \Omega_\theta} \left\| \frac{1}{N} \sum_{n=1}^{N} (w_n - 1) h_n(\theta) \right\|_1 \leq \delta \,.$$

When removing a set of $k$ points (i.e., $w = \mathbf{1} - \mathbf{1}_T$), the deviation includes $k$ terms of magnitude $\|h_i(\theta)\|_1 \approx d^2$, resulting in

$$\left\| \frac{1}{n} \sum (w_i - 1) h_i(\theta) \right\|_1 \approx \frac{kd^2}{n} \,.$$

The bound in Theorem B.2 requires this to be at most $\Delta_\delta = O(1)$, so we obtain the constraint:

$$\frac{kd^2}{n} \lesssim 1 \quad \Rightarrow \quad k \lesssim \frac{n}{d^2} \,.$$

This represents the main constraint required for Theorem B.2 to apply.

Finally, recall that in the main result of Theorem B.2, the error is bounded by

$$\mathrm{Err}_{\mathrm{IJ}} = \left\| \hat{\theta}_{\mathrm{IJ}}(w) - \hat{\theta}(w) \right\|_2 \lesssim C_{\mathrm{op}}^2 C_{\mathrm{IJ}} \delta^2 \,.$$

Given $\delta \approx \frac{kd^2}{n}$, and $C_{\mathrm{op}} = C_{\mathrm{IJ}} = O(1)$, we conclude:

$$\mathrm{Err}_{\mathrm{IJ}} \lesssim \left( \frac{kd^2}{n} \right)^2 = \frac{k^2 d^4}{n^2} \,.$$

## C   Experimental Details

We based our experimental design on that of Koh et al. [KATL19] who evaluate standard influence functions in a similar setting in order to have a clearer benchmark for comparison.

### C.1   Model Training

We fit all the logistic regression models using the `scipy.optimize.minimize` function to train the model using `L-BFGS-B`, and set a very strict stopping criterion to ensure that we converge to the global optimum and suppress dependencies on the initial weights when using a warm-start retrain.

For the DogFish and Enron datasets also considered by Koh et al., we used the same $L_2$ regularization parameter, and for all new datasets, we set the regularization to $1E - 5$.

### C.2   Removal Set Construction

Similar to Koh et al., we evaluate our data attribution methods on removals of "correlated" sets of samples from every regression. We focus on relatively fewer sample removals, varying the number of samples linearly along the range from $0.1\%$ to $5\%$ of the training set. For each dataset and each group construction strategy, we select 40 such sets of samples (1 for each size).

For each such size $k$, we construct removal sets of size $k$ using the following strategies

1. Clustered Samples: we construct sets of samples clustered either by a single feature or by $L_2$ distance. When clustering by a single feature, for each set of samples to remove, we select a random sample $i \in [n]$ and a random feature $j \in [d]$, and output the $k$ samples for which $X_{i',j}$ is closest to $X_{i,j}$. When clustering by $L_2$ distance, we select the center sample $i \in [n]$ uniformly at random and output the $k$ samples closest to it in $L_2$ norm.

2. Top Percentile Samples: For each of the metrics, we construct a top-percentile set of samples of size $k$, by selecting first selecting the top $2k$ samples and outputting a random subset of half of them. We consider the metrics of: high positive / negative influence on test loss and high positive / negative influence on test predictions, both computed using the standard influence function to keep our benchmark comparable with that of Koh et al.

3. Random Subsets: $k$ samples selected uniformly at random.

## C.3 Datasets and Embeddings

We consider several classification tasks in this paper. For each, we extract features from a particular modality (vision, NLP, or audio), embed them into a $d$-dimensional representation using a frozen pretrained model, and train a logistic regression classifier on a relevant 2-class classification problem.

For the Enron and DogFish datasets, we try to keep to the same conventions as Koh et al. [KATL19] for a clean comparison.

**ESC-50 embedded using OpenL3**   ESC-50 is a dataset of $\approx 5$ second audio clips each corresponding to one of $50$ categories with $40$ samples from each category [Pic15]. We convert this to a 2 class classification problem by dividing the categories into "natural" sounds (*breathing*, *cat*, *cow*, etc.) and "artificial" sounds (*airplane*, *chainsaw*, *clapping* etc.).

We embed these audio samples using last-layer embeddings of the OpenL3 python library [CWSB19]. This produces $d = 512$ dimensional embeddings, and we separate them into train and test samples using a random $80 - 20$ train-test split.

**CIFAR-2 embedded using ResNet-50**   We consider 2 CIFAR-2 datasets generated by limiting the CIFAR-10 dataset [Kri09] to 2 classes (Cat vs Dog, and Automobile vs Truck).

The photos from both train and test sets are embedded using the last-layer embeddings of the default pretrained ResNet-50 model in the `torchvision` python library [Tor16].

**DogFish embedded with Inception v3**   We reproduce the DogFish dataset from Koh et al. [KATL19].

This dataset contains photos of dogs and fish from the ImageNet dataset [RDS$^+$15] embedded using frozen last-layer embeddings of the Inception v3 network [SVI$^+$16].

**Enron embedded with Spacy**   We reproduce the Enron dataset from Koh et al. [KATL19].

This NLP dataset consists of Spam vs Ham emails [MAP06] embedded using a bag-of-words embedding with the `spacy` python library using the "en_core_web_sm" dictionary. We note that our embeddings for the Enron dataset may differ slightly from those of Koh et al. [KATL19], likely due to version differences in the `spacy` library. However, our empirical results are consistent with theirs.

**IMDB embedded with BERT**   We consider the NLP IMDB sentiment analysis dataset consisting of 50000 movie reviews classified into *positive* and *negative* [MDP$^+$11]. We embed the text reviews using the BERT model [DCLT19].

## C.4 Experiments

An implementation of our experiments is available at github.com/ittai-rubinstein/rescaled-influence-functions. This appendix provides a concise overview of the procedures implemented in the accompanying code.

### C.4.1 Comparison of Influence and Actual Effect

To produce Figure 1, we select sets of samples to remove based on the methods described in Appendix C.2. For each set of samples we retrain the logistic regression model without these samples to obtain the ground truth effect on the change in the metric $f$, and compare to the application of the same metric $f$ to the models predicted by each of the data attribution techniques.

**Removal effect vs influence**   One minor distinction considered in the appendix of Koh et al. [KATL19] is between the influence on a metric and the "parameter influence" on a metric. They define the influence on a metric to be the inner product between the gradient of the metric and the estimated change in model parameters

$$I_{f,w}^{\text{inf}} = \langle \nabla f, \boldsymbol{\theta}_w^{\text{inf}} - \boldsymbol{\theta} \rangle,$$

and the parameter influence of a set of removals (which we simply call the "removal effect") to be

$$I_{f,w}^{\text{param. inf.}} = f\left(\boldsymbol{\theta}_w^{\text{inf}}\right) - f\left(\boldsymbol{\theta}\right).$$

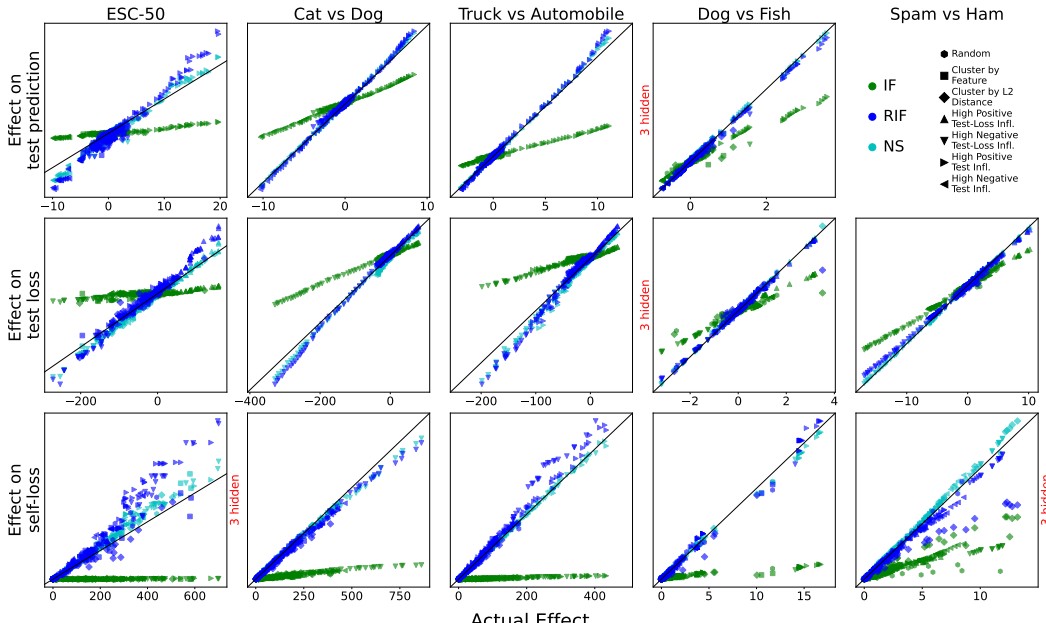

Figure 4: Accuracy of IF versus RIF compared across datasets from image classification (DogFish, Cat vs Dog, Truck vs Automobile), natural language (Spam vs Ham), and audio (ESC-50). Each data-point in this experiment is generated as its equivalent in Figure 1, except that instead of evaluating the metric $f$ (e.g., test-loss) on the retrained model or the data model prediction of the retrain effect, we use the leading order Taylor approximation of the change in this metric. There is no major qualitative difference between the results of this experiment and the ones reported in Figure 1, so we decided to keep the original evaluation for a clearer apples-to-apples comparison.

We use the latter method to produce all the data points in Figures 1 and 2 (the metric considered in Figure 3 is linear so it is not affected by this distinction). However, similar to Koh et al., we observe very little effect to using the linear method instead.

### C.4.2   Varying $n$ and $\lambda$

In these experiments we repeated the same experimental procedure as the one used to generate Figure 1, but with varying levels of $L_2$ regularization for the DogFish dataset and subsampling the IMDB dataset to different numbers of samples (via uniformly random draws). We report the effect of these removals on self-loss.

### C.4.3   Data Poisoning

To ground our results we consider a particular application of data attribution for detecting data poisoning attacks. We consider the simple data poisoning attack, where an adversary trying to flip our models prediction on some test sample (selected uniformly at random) and adds this sample with a flipped label to the train set. We then run IF and RIF data attributions on the poisoned dataset and use them to predict the effect of the poisoned sample on its own logit ($z_i = \langle \boldsymbol{\theta}, \mathbf{x}_i \rangle$) and compare this to the ground truth of a full retrain.

### C.5   Licensing of External Assets

We summarize the license information for all datasets and pretrained models used in our experiments. All assets are cited in the main text.

### Notes

Assets without explicit licenses (e.g., CIFAR-10, Enron, IMDB) are used strictly for non-commercial research purposes. We do not redistribute any datasets or pretrained weights.

Table 3: License summary for datasets used in our experiments. All assets are cited and used in accordance with their respective terms.

| Asset | Source | License | Use / Notes |
|---|---|---|---|
| ESC-50 | [Pic15] | CC BY-NC 3.0 | Freely available for non-commercial research use |
| CIFAR-10 | [Kri09] | Not specified | Widely used in academic settings; original authors affiliated with the University of Toronto |
| ImageNet | [RDS+15] | Custom terms | Access requires agreement to ImageNet's non-commercial license |
| Enron Spam | [MAP06] | Not specified | Used under standard academic fair use; available via public research repositories |
| IMDB Reviews | [MDP+11] | Not specified | Publicly downloadable from the Stanford AI Lab; used for academic research |

Table 4: License summary for pretrained models and libraries. All tools are used under compatible terms for non-commercial research.

| Model | Version | License | Use / Notes |
|---|---|---|---|
| OpenL3 | v0.4.2 | MIT | Permissive open-source license; commercial use allowed |
| ResNet-50 (TorchVision) | v0.13.1 | BSD 3-Clause | Standard pretrained model from TorchVision; license is permissive, but pretrained weights originate from ImageNet |
| Inception v3 | — | Apache 2.0 | Model license is permissive; weights trained on ImageNet, which restricts downstream use |
| spaCy | v3.8.2 | MIT | Freely usable model provided by spaCy; license allows commercial and academic use |
| BERT (Transformers) | bert-base-uncased (v4.36.2) | Apache 2.0 | Hugging Face model with a permissive license; trained on BookCorpus and Wikipedia, which may have unclear redistribution terms |

