# OpenReview forum: "Rescaled Influence Functions: Accurate Data Attribution in High Dimension"
_NeurIPS.cc/2025/Conference — NeurIPS 2025 poster_

### Official Review · Reviewer_CdeV · 2025-06-29

**Clarity:** 4
**Significance:** 3
**Originality:** 3
**Rating:** 5
**Confidence:** 5

**Summary:**

The paper presents rescaled influence functions, which make use of per-sample Hessians to rescale influence functions and achieve improved accuracy in capturing true training data attribution, especially for group influence, while preserving the linearity of group influence.

**Questions:**

- How strongly would a poor estimation of the Hessian affect the error in RIF compared to IF? Would the rescaling through poor leverage scores increase the error?
- How stable are RIFs wrt the randomness from the training process in non-convex models? Do you maybe have some insights from your experiments with e.g. ResNet50?

**Ethical Concerns:**

["NO or VERY MINOR ethics concerns only"]

**Final Justification:**

I recommend accepting this paper because it is a well-written paper proposing a simple and principled approach to improve influence functions that has yielded convincing empirical results.

**Limitations:**

Yes

**Quality:**

4

**Strengths And Weaknesses:**

Strengths:
- The paper presents a theoretically-grounded improvement to influence functions and underlines their analysis with empirical evidence with experiments with different model types and data modalities.
- The proposed method is simple to implement and therefore likely easy to adopt.

Weaknesses:
- Influence functions are known to be fragile in deep learning [a], which limits their applicability. The paper does not present a sensitivity analysis of RIF to test whether they are also fragile or improve IF also in this regard.

[a] Basu et al. (2020): Influence Functions in deep learning are fragile.

---

> ### Author Rebuttal · Authors · 2025-07-30
>
> Thank you for the detailed review and feedback!
>
> **Reviewer Questions**
>
> **Question 1:** This depends on how the Hessian is approximated. Common approaches to efficiently estimating the Hessian of deeper neural nets (e.g., as used in TRAK - Park et al. 2023), is to use a combination of K-FAC (to avoid having to directly differentiate the model to a second order) and Random Projection to a lower dimensional embedding (for faster linear algebra). Below, we expand on each of these methods and why we expect them to work well within the framework of RIF.
>
> **Question 2:** We have not yet run a comprehensive set of experiments into this question. Our experience so far supports the intuition detailed below (see Ensembling and Non-Convex Models) that RIF produces a good estimate on how each local minimum is shifted by sample removals and that the distribution of local minima approached by the model is fairly robust under sample removals, allowing us to capture the overall drift of the population of local minima by using an ensemble average RIF over a large number of initial models.
>
>
> **Larger Models, Approximate Hessians and Efficient Influence Functions**
>
> One of the challenges in applying influence functions to larger models is their significant computational overhead, particularly due to the need to compute and invert the Hessian of the loss. Several approaches have been considered for estimating influences without inverting a full loss Hessian, but we will focus on the random projection approach used in TRAK (Park et al. 2023), which is an IF-based data attribution method. We describe how the random projection approach combines cleanly with RIF to approximate the leverage scores used in RIF without the computational cost of a full Hessian inversion. While random projections lead to some loss in accuracy compared to full Hessian inversion, we believe that this loss is unlikely to be any worse for RIF-based data attribution than for IF-based data attribution.
>
> In this approach, instead of computing $\langle \text{IF}, v\rangle = H^{-1} g$, we select a random projection $\Pi :\mathbb{R}^d \to \mathbb{R}^{e \ll d}$ and estimate the model change in this projected space
> $$ \langle \text{IF}_{\text{TRAK}}, v\rangle \coloneqq v^T \Pi^T (\Pi H \Pi^T)^{-1} \Pi g \approx C \cdot \langle \text{IF}, v\rangle $$
>
> This approximation is supported by the Johnson-Lindenstrauss lemma, which states that with high probability, random projections scale down inner products by the ratio of dimensions. In other words, for any fixed vectors $v, w \in \mathbb{R}^d$, and a random projection $\Pi$ that outputs vectors of embedded-dimension $e$, with high probability
> $$ \langle \Pi v, \Pi w\rangle \approx \frac{e}{d} \langle v,w\rangle $$
>
> Therefore, in order to capture the correct scaling with a TRAK-inspired version of RIF, for a model with rank-1 Hessian updates (see below why this should suffice even for deeper models), we could use an approximation of the form
> $$\langle v, \text{RIF}\rangle \approx \langle v, \text{RIF}_{\text{projected}}\rangle \coloneqq \frac{d}{e} \frac{\langle \Pi v, (\Pi H \Pi^T)^{-1} \Pi g_i \rangle}{1 - L_i}$$
> where $L_i = \frac{d}{e} \beta_i v_i^T \Pi^T (\Pi H \Pi^T)^{-1} \Pi v_i$ approximates the $i$th leverage.
>
> Due to the Johnson Lindenstrauss lemma, we expect this method to produce an accurate estimate of the exact RIF.
>
> **Ensembling and Non-Convex Models**
>
> Another major challenge with using influence functions for deeper neural nets is the fact that loss minimization for deep neural networks is a nonconvex problem. Therefore, each training run will approach a different local minimum.
>
> Inspired by TRAK, we think of the influence function as estimating the shift of a single local minimum due to the removal of a set of samples. Instead of the influence function evaluated at a single trained model, TRAK uses an ensemble of trained models and averages the influence function estimates of their change under sample removal. This can be viewed as an estimate of the overall drift of the local minima.
>
> We expect that plugging in RIF rather than IF to the TRAK approach should yield a better estimate of the overall drift. This is because, while we expect influence functions to capture the direction of change of each local minima, we do not expect it to capture the scale of this change. By contrast, just as in the convex setting, we expect that RIF accurately captures both the direction and magnitude of the change in the location of a local minimum when a sample is removed from the training set.
>
> **K-FAC and Low-Rank Hessian Updates**
> Another observation which we did not mention in the paper is that the low-rank inverse-Hessian updates needed for RIF can be evaluated efficiently via Sherman-Morrison even for deeper neural nets provided they use ReLU activation. This is an important factor in ensuring that RIF can be evaluated with little overhead compared to IF even beyond logistic regression.
>
> This is because Neural Networks with ReLU activations are piece-wise linear as a function of the network input, meaning that, locally, each sample's contribution to the Hessian is exactly equal to its approximation to the Hessian of its NTK-based logistic regression (i.e., the K-FAC approximation is exact in this sense).
>
> For deeper neural networks with other activation functions, the rank of the Hessian update would be at most equal to the depth of the neural network. However, we expect that a rescaling based on the rank-1 Hessian update corresponding to the NTK approximation would be a good approximation to the leave-one-out Hessian.
>
> All of these updates are rank-1, allowing for fast evaluation using the Sherman-Morrison formula and using the low-rank embedding procedure described above.

---

> > ### Comment · Reviewer_CdeV · 2025-08-04
> >
> > Thank you for your detailed response. I have a follow up question to question 2: Did you conduct your experiments across multiple random seeds of retraining? If yes, it would be interesting to compare the stability of influence scores using IF and RIF.

---

> > > ### Author Response · Authors · 2025-08-07
> > >
> > > Thanks for the question!
> > > Yes, we ran all of our experiments with multiple different random seeds and saw the same qualitative behavior of IF vs RIF.

---

### Official Review · Reviewer_qech · 2025-07-01

**Clarity:** 4
**Significance:** 3
**Originality:** 3
**Rating:** 5
**Confidence:** 2

**Summary:**

Summary: This paper introduces a new data attribution method, Rescaled Influence Functions (RIF), that aims to more accurately estimate the effects of sample removal over the original IF method, and more efficient to compute than Newton Step (NS) by performing NS approximation.

The paper demonstrated empirically that, especially under overparameterized settings, RIF is able to estimate the leave-T-out effect at a similar accuracy of NS while offering significantly less computational overhead.

**Questions:**

Questions:

1. Wouldn’t it be clearer to present the empirical results (Section 2) after introducing the assumptions and theorems (Section 3)?

2. Could the authors suggest a few potential future research directions to build on the proposed RIF?

**Ethical Concerns:**

["NO or VERY MINOR ethics concerns only"]

**Final Justification:**

Questions and concerns have been addressed / answered accordingly.

**Limitations:**

yes

**Quality:**

3

**Strengths And Weaknesses:**

Strengths:

1. Strong empirical results and thorough evaluation. Evaluated against many different settings and RIF seems to have consistently outperformed over standard IF.
2. Provides a theoretical bound on the error between RIF and NS.

Weakness:

1. While the paper mentions that RIF has a computational overhead compared to standard IF, this aspect is not measured in detail beyond Section 1. It would be helpful to see empirical results showing the computational cost of each method on different datasets, similar to Figure 1 for accuracy.

---

> ### Author Rebuttal · Authors · 2025-07-30
>
> Thank you for the detailed review and feedback!
>
> The goal of our paper is to explore the effect of the rescaling phenomenon which leads to rescaled influence functions (RIF), so we decided to aim for the simplest setting where it could be evaluated. Inspired by Koh et al. 2019, we focus on logistic regressions, since they are easy to train and their convexity allows us to evaluate removal effects independent of hyperparameter tuning, while still providing an insight into the behavior of more complex models.
>
> In general, influence function (IF) data attribution is imperfect and has several flaws – especially in very high-dimensional and non-convex settings. RIF helps address some of these issues but we do not expect it to fix all of them. Below we detail how the ideas behind RIF could be used in combination with techniques such as the random projections and ensembling methods proposed in TRAK (Park et al. 2023), itself an adaptation of IF to very large non-convex models, to reduce the computational cost associated with large Hessians and improve accuracy in non-convex settings.
>
> While our initial experiments support the intuitions below, the goal of this paper is to evaluate the effect of rescaling in a simple scenario where it can be accurately measured. We believe that extensions to models so large that accurate Hessian inversion is infeasible, or to non-convex models, are beyond the scope of this paper, though they are natural and important directions for future work. We outline below some ideas regarding how such extensions could work not as a new contribution, but to answer the reviewer’s question regarding the applicability of our results beyond the setting considered in our paper.
>
> **Reviewer Questions**
>
> **Weakness 1.** Thank you for pointing out the lack of clear comparison of computational costs in our experiments. Below we list the wall-clock runtimes of the base model training, influence function computation and rescaling required for the main experiments of the paper. In all of these cases rescaling accounted for less than 1% of the total runtime. We plan to add a similar table to the camera ready version of the paper should it be accepted.
>
> **Question 1.** We chose this order because we feel that the empirical results are easy to interpret and convincing.
>
> **Question 2.** Certainly. One natural direction, discussed in detail below, is to apply RIFs to improve state-of-the-art data attribution in larger and non-convex models (e.g. neural nets). Another is to use RIF, or the ideas behind it, to improve on other existing uses of influence functions, such as for leave-k-out cross-validation and approximation of data Shapley values.
>
> **Larger Models, Approximate Hessians and Efficient Influence Functions**
> One of the challenges in applying influence functions to larger models is their significant computational overhead, particularly due to the need to compute and invert the Hessian of the loss. Several approaches have been considered for estimating influences without inverting a full loss Hessian, but we will focus on the random projection approach used in TRAK (Park et al. 2023), which is an IF-based data attribution method. We describe how the random projection approach combines cleanly with RIF to approximate the leverage scores used in RIF without the computational cost of a full Hessian inversion. While random projections lead to some loss in accuracy compared to full Hessian inversion, we believe that this loss is unlikely to be any worse for RIF-based data attribution than for IF-based data attribution.
>
> In this approach, instead of computing $\langle \text{IF}, v\rangle = H^{-1} g$, we select a random projection $\Pi :\mathbb{R}^d \to \mathbb{R}^{e \ll d}$ and estimate the model change in this projected space
> $$ \langle \text{IF}_{\text{TRAK}}, v\rangle \coloneqq v^T \Pi^T (\Pi H \Pi^T)^{-1} \Pi g \approx C \cdot \langle \text{IF}, v\rangle $$
>
> This approximation is supported by the Johnson-Lindenstrauss lemma, which states that with high probability, random projections scale down inner products by the ratio of dimensions. In other words, for any fixed vectors $v, w \in \mathbb{R}^d$, and a random projection $\Pi$ that outputs vectors of embedded-dimension $e$, with high probability
> $$ \langle \Pi v, \Pi w\rangle \approx \frac{e}{d} \langle v,w\rangle $$
>
> Therefore, in order to capture the correct scaling with a TRAK-inspired version of RIF, for a model with rank-1 Hessian updates (see below why this should suffice even for deeper models), we could use an approximation of the form
> $$\langle v, \text{RIF}\rangle \approx \langle v, \text{RIF}_{\text{projected}}\rangle \coloneqq \frac{d}{e} \frac{\langle \Pi v, (\Pi H \Pi^T)^{-1} \Pi g_i \rangle}{1 - L_i}$$
> where $L_i = \frac{d}{e} \beta_i v_i^T \Pi^T (\Pi H \Pi^T)^{-1} \Pi v_i$ approximates the $i$th leverage.
>
> Due to the Johnson Lindenstrauss lemma, we expect this method to produce an accurate estimate of the exact RIF.
>
> **Ensembling and Non-Convex Models**
> Another major challenge with using influence functions for deeper neural nets is the fact that loss minimization for deep neural networks is a nonconvex problem. Therefore, each training run will approach a different local minimum.
>
> Inspired by TRAK, we think of the influence function as estimating the shift of a single local minimum due to the removal of a set of samples. Instead of the influence function evaluated at a single trained model, TRAK uses an ensemble of trained models and averages the influence function estimates of their change under sample removal. This can be viewed as an estimate of the overall drift of the local minima.
>
> We expect that plugging in RIF rather than IF to the TRAK approach should yield a better estimate of the overall drift. This is because, while we expect influence functions to capture the direction of change of each local minima, we do not expect it to capture the scale of this change. By contrast, just as in the convex setting, we expect that RIF accurately captures both the direction and magnitude of the change in the location of a local minimum when a sample is removed from the training set.
>
> **K-FAC and Low-Rank Hessian Updates**
> Another observation which we did not mention in the paper is that the low-rank inverse-Hessian updates needed for RIF can be evaluated efficiently via Sherman-Morrison even for deeper neural nets provided they use ReLU activation. This is an important factor in ensuring that RIF can be evaluated with little overhead compared to IF even beyond logistic regression.
>
> This is because Neural Networks with ReLU activations are piece-wise linear as a function of the network input, meaning that, locally, each sample's contribution to the Hessian is exactly equal to its approximation to the Hessian of its NTK-based logistic regression (i.e., the K-FAC approximation is exact in this sense).
>
> For deeper neural networks with other activation functions, the rank of the Hessian update would be at most equal to the depth of the neural network. However, we expect that a rescaling based on the rank-1 Hessian update corresponding to the NTK approximation would be a good approximation to the leave-one-out Hessian.
>
> All of these updates are rank-1, allowing for fast evaluation using the Sherman-Morrison formula and using the low-rank embedding procedure described above.
>
> **Comparison of Runtimes**
>
> | Dataset      | Training | Hessian | Inversion | Influence | Rescaling        |
> |--------------|----------|---------|-----------|-----------|------------------|
> | ESC50        | 1.8 s    | 0.056 s | 0.0005 s  | 0.051 s   | 0.0033 s (0.2%)  |
> | CatDog       | 76 s     | 4.9 s   | 0.010 s   | 4.8 s     | 0.087 s (0.1%)   |
> | AutoTruck    | 48 s     | 4.9 s   | 0.0094 s  | 4.8 s     | 0.087 s (0.2%)   |
> | DogFish      | 0.43 s   | 0.92 s  | 0.0095 s  | 0.89 s    | 0.015 s (0.7%)   |
> | Enron        | 6.7 s    | 15 s    | 0.065 s   | 15 s      | 0.095 s (0.3%)   |
> | IMDB ($n=16d$) | 20 s     | 0.92 s  | 0.0012 s  | 0.87 s    | 0.044 s (0.2%)   |

---

> > ### Comment · Reviewer_qech · 2025-08-02
> >
> > I thank the authors for their thorough and thoughtful response. All of my questions have been addressed, and I have adjusted my score accordingly.

---

### Official Review · Reviewer_qLFf · 2025-07-02

**Clarity:** 3
**Significance:** 3
**Originality:** 3
**Rating:** 5
**Confidence:** 3

**Summary:**

This paper proposes a new method for influence functions that leverages Newton steps to compute the influence of individual samples, and then additively aggregates them to evaluate the influence of a subset. The method, called RIF, demonstrates improved performance over standard influence functions (IF) in the context of logistic regression across various scenarios, showing gains in both prediction accuracy and loss values.
RIF is also more effective in mitigating the impact of poisoned samples.

These improvements, however, come at the cost of additional computational overhead due to the need to compute the inverse Hessian matrix for each individual point.

**Questions:**

Mentioned in the weaknesses.

**Ethical Concerns:**

["NO or VERY MINOR ethics concerns only"]

**Final Justification:**

I think a borderline accept is an appropriate recommendation, as I am not sufficiently expert in this specific area to confidently recommend acceptance.

**Limitations:**

Yes

**Paper Formatting Concerns:**

Follows the guidelines

**Quality:**

3

**Strengths And Weaknesses:**

Strengths:
The paper considers a variety of scenarios and employs multiple metrics to evaluate accuracy and compare against standard Influence Functions (IF).
The theoretical development appears sound, correct, and thorough.
The additional computational overhead introduced by RIF seems justified given the observed improvements.

Weaknesses:
The main novelty of the paper—using Newton steps only for individual samples and then additively computing subset influence—is not particularly strong.
Moreover, the focus on logistic regression is somewhat narrow. Could the authors provide experimental results on more complex models? It would also be useful to compare IF and RIF in settings where influence is approximated, which is the typical use case of influence functions.

---

> ### Author Rebuttal · Authors · 2025-07-30
>
> Thank you for the detailed review and feedback!
>
> The goal of our paper is to explore the effect of the rescaling phenomenon which leads to rescaled influence functions (RIF), so we decided to aim for the simplest setting where it could be evaluated. Inspired by Koh et al. 2019, we focus on logistic regressions, since they are easy to train and their convexity allows us to evaluate removal effects independent of hyperparameter tuning, while still providing an insight into the behavior of more complex models.
>
> In general, influence function (IF) data attribution is imperfect and has several flaws – especially in very high-dimensional and non-convex settings. RIF helps address some of these issues but we do not expect it to fix all of them. Below we detail how the ideas behind RIF could be used in combination with techniques such as the random projections and ensembling methods proposed in TRAK (Park et al. 2023), itself an adaptation of IF to very large non-convex models, to reduce the computational cost associated with large Hessians and improve accuracy in non-convex settings.
>
> While our initial experiments support the intuitions below, the goal of this paper is to evaluate the effect of rescaling in a simple scenario where it can be accurately measured. We believe that extensions to models so large that accurate Hessian inversion is infeasible, or to non-convex models, are beyond the scope of this paper, though they are natural and important directions for future work. We outline below some ideas regarding how such extensions could work not as a new contribution, but to answer the reviewer’s question regarding the applicability of our results beyond the setting considered in our paper.
>
>
> **Reviewer Questions**
> We discuss below why we choose to focus on logistic regression. Regarding experiments with more complex models: while new results are out of scope for this rebuttal (see email from NeurIPS PC chairs with subject “Notes about rebuttal format”, which requests rebuttals discussing the current merits of the paper), we discuss below several of the key difficulties we would face in applying RIF to larger and more complex models below, and why we expect them to be surmountable.
>
>
> **Larger Models, Approximate Hessians and Efficient Influence Functions**
> One of the challenges in applying influence functions to larger models is their significant computational overhead, particularly due to the need to compute and invert the Hessian of the loss. Several approaches have been considered for estimating influences without inverting a full loss Hessian, but we will focus on the random projection approach used in TRAK (Park et al. 2023), which is an IF-based data attribution method. We describe how the random projection approach combines cleanly with RIF to approximate the leverage scores used in RIF without the computational cost of a full Hessian inversion. While random projections lead to some loss in accuracy compared to full Hessian inversion, we believe that this loss is unlikely to be any worse for RIF-based data attribution than for IF-based data attribution.
>
> In this approach, instead of computing $\langle \text{IF}, v\rangle = H^{-1} g$, we select a random projection $\Pi :\mathbb{R}^d \to \mathbb{R}^{e \ll d}$ and estimate the model change in this projected space
> $$ \langle \text{IF}_{\text{TRAK}}, v\rangle \coloneqq v^T \Pi^T (\Pi H \Pi^T)^{-1} \Pi g \approx C \cdot \langle \text{IF}, v\rangle $$
>
> This approximation is supported by the Johnson-Lindenstrauss lemma, which states that with high probability, random projections scale down inner products by the ratio of dimensions. In other words, for any fixed vectors $v, w \in \mathbb{R}^d$, and a random projection $\Pi$ that outputs vectors of embedded-dimension $e$, with high probability
> $$ \langle \Pi v, \Pi w\rangle \approx \frac{e}{d} \langle v,w\rangle $$
>
> Therefore, in order to capture the correct scaling with a TRAK-inspired version of RIF, for a model with rank-1 Hessian updates (see below why this should suffice even for deeper models), we could use an approximation of the form
> $$\langle v, \text{RIF}\rangle \approx \langle v, \text{RIF}_{\text{projected}}\rangle \coloneqq \frac{d}{e} \frac{\langle \Pi v, (\Pi H \Pi^T)^{-1} \Pi g_i \rangle}{1 - L_i}$$
> where $L_i = \frac{d}{e} \beta_i v_i^T \Pi^T (\Pi H \Pi^T)^{-1} \Pi v_i$ approximates the $i$th leverage.
>
> Due to the Johnson Lindenstrauss lemma, we expect this method to produce an accurate estimate of the exact RIF.
>
> **Ensembling and Non-Convex Models**
> Another major challenge with using influence functions for deeper neural nets is the fact that loss minimization for deep neural networks is a nonconvex problem. Therefore, each training run will approach a different local minimum.
>
> Inspired by TRAK, we think of the influence function as estimating the shift of a single local minimum due to the removal of a set of samples. Instead of the influence function evaluated at a single trained model, TRAK uses an ensemble of trained models and averages the influence function estimates of their change under sample removal. This can be viewed as an estimate of the overall drift of the local minima.
>
> We expect that plugging in RIF rather than IF to the TRAK approach should yield a better estimate of the overall drift. This is because, while we expect influence functions to capture the direction of change of each local minima, we do not expect it to capture the scale of this change. By contrast, just as in the convex setting, we expect that RIF accurately captures both the direction and magnitude of the change in the location of a local minimum when a sample is removed from the training set.
>
> **K-FAC and Low-Rank Hessian Updates**
> Another observation which we did not mention in the paper is that the low-rank inverse-Hessian updates needed for RIF can be evaluated efficiently via Sherman-Morrison even for deeper neural nets provided they use ReLU activation. This is an important factor in ensuring that RIF can be evaluated with little overhead compared to IF even beyond logistic regression.
>
> This is because Neural Networks with ReLU activations are piece-wise linear as a function of the network input, meaning that, locally, each sample's contribution to the Hessian is exactly equal to its approximation to the Hessian of its NTK-based logistic regression (i.e., the K-FAC approximation is exact in this sense).
>
> For deeper neural networks with other activation functions, the rank of the Hessian update would be at most equal to the depth of the neural network. However, we expect that a rescaling based on the rank-1 Hessian update corresponding to the NTK approximation would be a good approximation to the leave-one-out Hessian.
>
> All of these updates are rank-1, allowing for fast evaluation using the Sherman-Morrison formula and using the low-rank embedding procedure described above.
>
> **Comparison of Runtimes**
>
> | Dataset      | Training | Hessian | Inversion | Influence | Rescaling        |
> |--------------|----------|---------|-----------|-----------|------------------|
> | ESC50        | 1.8 s    | 0.056 s | 0.0005 s  | 0.051 s   | 0.0033 s (0.2%)  |
> | CatDog       | 76 s     | 4.9 s   | 0.010 s   | 4.8 s     | 0.087 s (0.1%)   |
> | AutoTruck    | 48 s     | 4.9 s   | 0.0094 s  | 4.8 s     | 0.087 s (0.2%)   |
> | DogFish      | 0.43 s   | 0.92 s  | 0.0095 s  | 0.89 s    | 0.015 s (0.7%)   |
> | Enron        | 6.7 s    | 15 s    | 0.065 s   | 15 s      | 0.095 s (0.3%)   |
> | IMDB ($n=16d$) | 20 s     | 0.92 s  | 0.0012 s  | 0.87 s    | 0.044 s (0.2%)   |

---

### Official Review · Reviewer_XBFo · 2025-07-03

**Clarity:** 3
**Significance:** 3
**Originality:** 3
**Rating:** 5
**Confidence:** 1

**Summary:**

The paper introduces Rescaled Influence Functions (RIF), a straightforward multiplicative re-weighting approach applied to classical influence-function scores to correct their tendency to underestimate leave-T-out effects in high-dimensional and over-parameterized models.

RIF maintains the attractive additivity and computational simplicity of first-order influence functions (IF), while achieving accuracy comparable to one-step Newton updates. Extensive experiments conducted across six diverse benchmarks—covering vision, audio, and NLP domains—demonstrate substantial accuracy improvements over IF, often by an order of magnitude. Additionally, a toy data-poisoning demonstration highlights the practical applicability of the method. The paper also provides theoretical insights supporting the method’s robustness and approximation quality in high-dimensional (low SNR) settings.

**Questions:**

See weaknesses, sorry dont have strong questions. Will read other reviews to form a better opinion.

**Ethical Concerns:**

["NO or VERY MINOR ethics concerns only"]

**Final Justification:**

All of my questions have been addressed, I maintain my score.

**Quality:**

3

**Strengths And Weaknesses:**

## Strengths

**Clarity:** The paper successfully motivates the setting, even for readers who may not be deeply familiar with the topic. The methodology is clear and approachable, and the results are presented in a well-structured manner.

**Novelty and Potential:** The paper addresses a well-known limitation of IFs within modern, over-parameterized models, offering an elegant and straightforward solution with substantial potential for various applications.

**Robust Empirical and Theoretical Support:** The empirical evaluation is extensive, covering image datasets (CIFAR-10, ImageNet), audio (ESC-50), and text (Enron spam). The variety of subset-selection strategies employed further strengthens the external validity of the results. Additionally, the theoretical analysis is well-developed and offers convincing guarantees.

## Weaknesses

Applications to Non-linear Functions: While the experiments are comprehensive within their chosen settings, the paper currently limits evaluation to logistic regression on frozen features. Could the authors clarify how the method might extend to more complex scenarios involving deep-network fine-tuning, large language models (LLMs), or Transformer-based vision architectures? This would help readers better understand potential limitations or challenges in scaling the method.

Comparison to other scalable methods: The primary theoretical results currently require convex losses and Hessians. Given the substantial computational challenges posed by billion-parameter models, such as difficulties with Hessian-vector products, could the authors elaborate on how other approaches like TRAK or FastIF perform?

Overall, the paper presents cool insights -- I really liked reading it and think it could be interesting to see at NeurIPS

---

> ### Author Rebuttal · Authors · 2025-07-30
>
> Thank you for the detailed review and feedback!
>
> The goal of our paper is to explore the effect of the rescaling phenomenon which leads to rescaled influence functions (RIF), so we decided to aim for the simplest setting where it could be evaluated. Inspired by Koh et al. 2019, we focus on logistic regressions, since they are easy to train and their convexity allows us to evaluate removal effects independent of hyperparameter tuning, while still providing an insight into the behavior of more complex models.
>
> In general, influence function (IF) data attribution is imperfect and has several flaws – especially in very high-dimensional and non-convex settings. RIF helps address some of these issues but we do not expect it to fix all of them. Below we detail how the ideas behind RIF could be used in combination with techniques such as the random projections and ensembling methods proposed in TRAK (Park et al. 2023), itself an adaptation of IF to very large non-convex models, to reduce the computational cost associated with large Hessians and improve accuracy in non-convex settings.
>
> While our initial experiments support the intuitions below, the goal of this paper is to evaluate the effect of rescaling in a simple scenario where it can be accurately measured. We believe that extensions to models so large that accurate Hessian inversion is infeasible, or to non-convex models, are beyond the scope of this paper, though they are natural and important directions for future work. We outline below some ideas regarding how such extensions could work not as a new contribution, but to answer the reviewer’s question regarding the applicability of our results beyond the setting considered in our paper.
>
> **Reviewer Questions**
> We discuss below how we expect RIF to apply to larger models with non-convex loss landscapes, although we leave thorough experimental investigation of this important question to future work. In particular, we address how to combine insights from TRAK with RIF to mitigate the computational burden of Hessian inversion, we discuss why the leave-one-out Hessian inverses needed by RIF remain relatively fast to compute in neural net settings, and we discuss why the idea from TRAK to handle non-convexity by averaging the IF over many training runs carries over to RIF.
>
>
> **Larger Models, Approximate Hessians and Efficient Influence Functions**
> One of the challenges in applying influence functions to larger models is their significant computational overhead, particularly due to the need to compute and invert the Hessian of the loss. Several approaches have been considered for estimating influences without inverting a full loss Hessian, but we will focus on the random projection approach used in TRAK (Park et al. 2023), which is an IF-based data attribution method. We describe how the random projection approach combines cleanly with RIF to approximate the leverage scores used in RIF without the computational cost of a full Hessian inversion. While random projections lead to some loss in accuracy compared to full Hessian inversion, we believe that this loss is unlikely to be any worse for RIF-based data attribution than for IF-based data attribution.
>
> In this approach, instead of computing $\langle \text{IF}, v\rangle = H^{-1} g$, we select a random projection $\Pi :\mathbb{R}^d \to \mathbb{R}^{e \ll d}$ and estimate the model change in this projected space
> $$ \langle \text{IF}_{\text{TRAK}}, v\rangle \coloneqq v^T \Pi^T (\Pi H \Pi^T)^{-1} \Pi g \approx C \cdot \langle \text{IF}, v\rangle $$
>
> This approximation is supported by the Johnson-Lindenstrauss lemma, which states that with high probability, random projections scale down inner products by the ratio of dimensions. In other words, for any fixed vectors $v, w \in \mathbb{R}^d$, and a random projection $\Pi$ that outputs vectors of embedded-dimension $e$, with high probability
> $$ \langle \Pi v, \Pi w\rangle \approx \frac{e}{d} \langle v,w\rangle $$
>
> Therefore, in order to capture the correct scaling with a TRAK-inspired version of RIF, for a model with rank-1 Hessian updates (see below why this should suffice even for deeper models), we could use an approximation of the form
> $$\langle v, \text{RIF}\rangle \approx \langle v, \text{RIF}_{\text{projected}}\rangle \coloneqq \frac{d}{e} \frac{\langle \Pi v, (\Pi H \Pi^T)^{-1} \Pi g_i \rangle}{1 - L_i}$$
> where $L_i = \frac{d}{e} \beta_i v_i^T \Pi^T (\Pi H \Pi^T)^{-1} \Pi v_i$ approximates the $i$th leverage.
>
> Due to the Johnson Lindenstrauss lemma, we expect this method to produce an accurate estimate of the exact RIF.
>
> **Ensembling and Non-Convex Models**
> Another major challenge with using influence functions for deeper neural nets is the fact that loss minimization for deep neural networks is a nonconvex problem. Therefore, each training run will approach a different local minimum.
>
> Inspired by TRAK, we think of the influence function as estimating the shift of a single local minimum due to the removal of a set of samples. Instead of the influence function evaluated at a single trained model, TRAK uses an ensemble of trained models and averages the influence function estimates of their change under sample removal. This can be viewed as an estimate of the overall drift of the local minima.
>
> We expect that plugging in RIF rather than IF to the TRAK approach should yield a better estimate of the overall drift. This is because, while we expect influence functions to capture the direction of change of each local minima, we do not expect it to capture the scale of this change. By contrast, just as in the convex setting, we expect that RIF accurately captures both the direction and magnitude of the change in the location of a local minimum when a sample is removed from the training set.
>
> **K-FAC and Low-Rank Hessian Updates**
> Another observation which we did not mention in the paper is that the low-rank inverse-Hessian updates needed for RIF can be evaluated efficiently via Sherman-Morrison even for deeper neural nets provided they use ReLU activation. This is an important factor in ensuring that RIF can be evaluated with little overhead compared to IF even beyond logistic regression.
>
> This is because Neural Networks with ReLU activations are piece-wise linear as a function of the network input, meaning that, locally, each sample's contribution to the Hessian is exactly equal to its approximation to the Hessian of its NTK-based logistic regression (i.e., the K-FAC approximation is exact in this sense).
>
> For deeper neural networks with other activation functions, the rank of the Hessian update would be at most equal to the depth of the neural network. However, we expect that a rescaling based on the rank-1 Hessian update corresponding to the NTK approximation would be a good approximation to the leave-one-out Hessian.
>
> All of these updates are rank-1, allowing for fast evaluation using the Sherman-Morrison formula and using the low-rank embedding procedure described above.
>
> **Comparison of Runtimes**
>
> | Dataset      | Training | Hessian | Inversion | Influence | Rescaling        |
> |--------------|----------|---------|-----------|-----------|------------------|
> | ESC50        | 1.8 s    | 0.056 s | 0.0005 s  | 0.051 s   | 0.0033 s (0.2%)  |
> | CatDog       | 76 s     | 4.9 s   | 0.010 s   | 4.8 s     | 0.087 s (0.1%)   |
> | AutoTruck    | 48 s     | 4.9 s   | 0.0094 s  | 4.8 s     | 0.087 s (0.2%)   |
> | DogFish      | 0.43 s   | 0.92 s  | 0.0095 s  | 0.89 s    | 0.015 s (0.7%)   |
> | Enron        | 6.7 s    | 15 s    | 0.065 s   | 15 s      | 0.095 s (0.3%)   |
> | IMDB ($n=16d$) | 20 s     | 0.92 s  | 0.0012 s  | 0.87 s    | 0.044 s (0.2%)   |

---

> > ### Comment · Reviewer_XBFo · 2025-08-07
> > **Acknowledgement**
> >
> > Thank you for your rebuttal. It satisfactorily addresses my concerns, I especially liked the timing comparison. I will maintain my positive score-- eager to see this at NeurIPS.

---

### Official Review · Reviewer_xHGk · 2025-07-03

**Clarity:** 3
**Significance:** 2
**Originality:** 2
**Rating:** 3
**Confidence:** 5

**Summary:**

This paper addresses the challenge of accurately estimating the impact of training data removal in high-dimensional machine learning models, where traditional influence functions (IFs) often underestimate the true effects due to their reliance on first-order approximations. The authors propose rescaled influence functions (RIF), a computationally efficient method that incorporates limited higher-order information by leveraging leave-one-out Hessian adjustments while retaining the additivity and scalability of IFs. RIF approximates the Newton step (NS) update—a more accurate but costly second-order approximation—by precomputing leave-one-out Hessian inverses, which can be efficiently derived via matrix inversion lemmas for low-rank updates. Theoretical analysis demonstrates that RIF achieves tighter bounds on prediction errors compared to IFs, particularly when the number of parameters approaches or exceeds the sample size. Experiments across diverse datasets (image, text, audio) confirm RIF’s superior accuracy in predicting leave-T-out effects on test predictions, losses, and self-loss metrics, especially under high-dimensional or weakly regularized settings.

**Questions:**

1. The theoretical analysis assumes convexity and smooth losses, yet the method is claimed to be model-agnostic. How does RIF perform in highly non-convex settings, such as deep neural networks with multiple local minima? For example, do the rescaling factors derived from leave-one-out Hessians remain stable across different initialization points or training trajectories?

2. The tradeoff between RIF and IF accuracy is shown to depend on regularization strength λ (Figure 2), but how sensitive is RIF to other hyperparameters (e.g., learning rate, batch size during training)? Could stochastic Hessian approximations (e.g., using mini-batch Hessians) destabilize RIF’s rescaling factors in practice?

3. While the paper claims negligible overhead over IFs via Sherman-Morrison updates, how does RIF scale for models with millions of parameters (e.g., transformers)? For instance, in settings where Hessian inversion is approximated via low-rank methods (e.g., Kronecker-factored approximations), does RIF’s accuracy degrade, and what is the wall-clock time/memory footprint?

If the author can effectively solve the above problems, I will improve my score.

**Ethical Concerns:**

["NO or VERY MINOR ethics concerns only"]

**Final Justification:**

The author's response fails to address the critical issues regarding RIF's dependence on hyperparameters (Question 2) and its lack of scalability to large models (e.g., LLMs, VLMs) (Question 3). Therefore, I consider this work to represent only a minor modification of IF, which reinforces my original recommendation for rejection. I have also increased my Confidence score in this decision. Notably, I observe that other reviewers leaning toward acceptance have relatively low Confidence scores in their evaluations.

**Limitations:**

Yes.

**Paper Formatting Concerns:**

No.

**Quality:**

3

**Strengths And Weaknesses:**

### Strengths
1. The paper addresses a critical limitation of traditional influence functions (IFs)—their inaccuracy in high-dimensional or overparameterized regimes. By introducing rescaled influence functions (RIFs), which incorporate limited second-order information via leave-one-out Hessian adjustments, the authors demonstrate significant accuracy gains over IFs. This is theoretically justified via matrix-perturbation analysis and validated empirically across diverse datasets (image, text, audio). The RIF framework retains the computational efficiency and additivity of IFs while achieving near-Newton-step accuracy, making it a practical drop-in replacement.

2. RIFs are model-agnostic and task-agnostic, applicable to any empirical risk minimization problem with smooth losses. The method’s ability to detect subtle data poisoning attacks that evade IF-based diagnostics highlights its robustness and utility in security-critical applications. The theoretical analysis also provides tight error bounds under realistic assumptions (e.g., cross-sample incoherence), ensuring reliability in settings where $ n \approx d $ or $ n \ll d $, which are common in modern ML.

3. The experiments span vision, NLP, and audio domains, demonstrating consistent improvements over IFs in predicting leave-T-out effects (test predictions, losses, self-loss). The results include ablation studies on regularization and dimensionality tradeoffs, reinforcing the method’s practicality. For instance, RIFs outperform IFs even in weakly regularized or high-dimensional scenarios (e.g., DogFish dataset), showcasing their versatility.

---

### Weaknesses
1. While the paper claims negligible overhead over IFs (e.g., via Sherman-Morrison updates for low-rank Hessians), the requirement to compute $ n $ leave-one-out Hessian inverses could become prohibitive for extremely large models or datasets. For example, in deep learning with millions of parameters, even low-rank approximations or matrix inversion lemmas may incur non-trivial memory and time costs, limiting RIF’s applicability to resource-constrained settings.

2. The theoretical guarantees rely on convexity and smooth loss functions, which may not hold for complex, non-convex models like deep neural networks. Although the authors mention extensions to non-convex settings, the empirical validation focuses on logistic regression (a convex problem). This raises questions about RIF’s performance in highly non-convex regimes, where Hessian approximations might be less reliable or require additional stabilization.

3. While the paper demonstrates RIF’s superiority in detecting synthetic poisoning attacks, it does not evaluate its effectiveness against advanced, real-world data poisoning strategies (e.g., clean-label poisoning or gradient-matching attacks). The experiments focus on simple mislabeling, leaving open whether RIF generalizes to more sophisticated threats. Additionally, the method’s sensitivity to hyperparameters (e.g., regularization strength $ \lambda $) is only partially explored, potentially affecting reproducibility in practice.

---

> ### Author Rebuttal · Authors · 2025-07-30
>
> Thank you for the detailed review and feedback!
>
> The goal of our paper is to explore the effect of the rescaling phenomenon which leads to rescaled influence functions (RIF), so we decided to aim for the simplest setting where it could be evaluated. Inspired by Koh et al. 2019, we focus on logistic regressions, since they are easy to train and their convexity allows us to evaluate removal effects independent of hyperparameter tuning, while still providing an insight into the behavior of more complex models.
>
> In general, influence function (IF) data attribution is imperfect and has several flaws – especially in very high-dimensional and non-convex settings. RIF helps address some of these issues but we do not expect it to fix all of them. Below we detail how the ideas behind RIF could be used in combination with techniques such as the random projections and ensembling methods proposed in TRAK (Park et al. 2023), itself an adaptation of IF to very large non-convex models, to reduce the computational cost associated with large Hessians and improve accuracy in non-convex settings.
>
> While our initial experiments support the intuitions below, the goal of this paper is to evaluate the effect of rescaling in a simple scenario where it can be accurately measured. We believe that extensions to models so large that accurate Hessian inversion is infeasible, or to non-convex models, are beyond the scope of this paper, though they are natural and important directions for future work. We outline below some ideas regarding how such extensions could work not as a new contribution, but to answer the reviewer’s question regarding the applicability of our results beyond the setting considered in our paper.
>
>
> **Question 1:** Our theoretical analysis focuses on the convex setting because there is very little that we know how to prove about the convergence of deep neural networks in general, and we do not expect to be able to prove any simple statement about which local minimum is reached when optimizing in a non-convex setting.
>
> However, we do believe that IF/RIF, applied to a trained model which sits at a local minimum of the loss landscape, would provide an estimate of how that particular local minimum of the landscape would shift under sample removals. For the same reasons as in the convex setting, we expect RIF to provide a more accurate estimate of this shift than IF. We can put together such estimates of the shift of a single local minimum in the same way as TRAK does for IF.
>
> More concretely, if we are willing to assume that:
>
> SGD converges to some distribution over these local minima
> When a small subset of samples are removed, each local minimum itself moves (but local minima do not disappear and new ones are not introduced), and
> When a small subset of samples are removed, the distribution over local minima reached by SGD does not change too much,
>
> then we can use IF, or even better RIF, to provide an estimate of how each of these local minimum is shifted by the sample removal, and then average this estimate over many local minima found via an ensemble of base models, to yield an estimate of the shift in the distribution of trained models (see the Ensembling section of our rebuttal). Moreover, we expect to be able to show this theoretically using essentially the same techniques as in Theorem 3.1.
>
> **Question 2:** This is a very good question. Indeed, Basu et al. 2020 show that the quality of predictions produced by influence functions are very sensitive to such hyper-parameters and ultimately, we expect RIF to have a similar dependence. This is why we chose to focus on logistic regressions, where the rescaling effect (the difference between IF and RIF) can be more easily isolated, and leave the question of the effects of hyper-parameters to future research.
>
> **Question 3:** We expect to be able to use rank-1 inverse-Hessian updates (which allow very fast Sherman-Morrison based evaluation) in most applications, or at worst low-rank updates (which can still be evaluated efficiently using the Woodbury matrix identity), and for our methods to translate well to the K-FAC approximation of the Hessian – see K-FAC section of our rebuttal.
>
> Thank you for pointing out the lack of clear comparison of computational costs in our experiments. Below we list the wall-clock runtimes of the base model training, influence function computation and rescaling required for the main experiments of the paper. In all of these cases rescaling accounted for less than 1% of the total runtime. We plan to add a similar table to the camera ready version of the paper should it be accepted.
>
>
> **Larger Models, Approximate Hessians and Efficient Influence Functions**
> One of the challenges in applying influence functions to larger models is their significant computational overhead, particularly due to the need to compute and invert the Hessian of the loss. Several approaches have been considered for estimating influences without inverting a full loss Hessian, but we will focus on the random projection approach used in TRAK (Park et al. 2023), which is an IF-based data attribution method. We describe how the random projection approach combines cleanly with RIF to approximate the leverage scores used in RIF without the computational cost of a full Hessian inversion. While random projections lead to some loss in accuracy compared to full Hessian inversion, we believe that this loss is unlikely to be any worse for RIF-based data attribution than for IF-based data attribution.
>
> In this approach, instead of computing $\langle \text{IF}, v\rangle = H^{-1} g$, we select a random projection $\Pi :\mathbb{R}^d \to \mathbb{R}^{e \ll d}$ and estimate the model change in this projected space
> $$ \langle \text{IF}_{\text{TRAK}}, v\rangle \coloneqq v^T \Pi^T (\Pi H \Pi^T)^{-1} \Pi g \approx C \cdot \langle \text{IF}, v\rangle $$
>
> This approximation is supported by the Johnson-Lindenstrauss lemma, which states that with high probability, random projections scale down inner products by the ratio of dimensions. In other words, for any fixed vectors $v, w \in \mathbb{R}^d$, and a random projection $\Pi$ that outputs vectors of embedded-dimension $e$, with high probability
> $$ \langle \Pi v, \Pi w\rangle \approx \frac{e}{d} \langle v,w\rangle $$
>
> Therefore, in order to capture the correct scaling with a TRAK-inspired version of RIF, for a model with rank-1 Hessian updates (see below why this should suffice even for deeper models), we could use an approximation of the form
> $$\langle v, \text{RIF}\rangle \approx \langle v, \text{RIF}_{\text{projected}}\rangle \coloneqq \frac{d}{e} \frac{\langle \Pi v, (\Pi H \Pi^T)^{-1} \Pi g_i \rangle}{1 - L_i}$$
> where $L_i = \frac{d}{e} \beta_i v_i^T \Pi^T (\Pi H \Pi^T)^{-1} \Pi v_i$ approximates the $i$th leverage.
>
> Due to the Johnson Lindenstrauss lemma, we expect this method to produce an accurate estimate of the exact RIF.
>
> **Ensembling and Non-Convex Models**
> Another major challenge with using influence functions for deeper neural nets is the fact that loss minimization for deep neural networks is a nonconvex problem. Therefore, each training run will approach a different local minimum.
>
> Inspired by TRAK, we think of the influence function as estimating the shift of a single local minimum due to the removal of a set of samples. Instead of the influence function evaluated at a single trained model, TRAK uses an ensemble of trained models and averages the influence function estimates of their change under sample removal. This can be viewed as an estimate of the overall drift of the local minima.
>
> We expect that plugging in RIF rather than IF to the TRAK approach should yield a better estimate of the overall drift. This is because, while we expect influence functions to capture the direction of change of each local minima, we do not expect it to capture the scale of this change. By contrast, just as in the convex setting, we expect that RIF accurately captures both the direction and magnitude of the change in the location of a local minimum when a sample is removed from the training set.
>
> **K-FAC and Low-Rank Hessian Updates**
> Another observation which we did not mention in the paper is that the low-rank inverse-Hessian updates needed for RIF can be evaluated efficiently via Sherman-Morrison even for deeper neural nets provided they use ReLU activation. This is an important factor in ensuring that RIF can be evaluated with little overhead compared to IF even beyond logistic regression.
>
> This is because Neural Networks with ReLU activations are piece-wise linear as a function of the network input, meaning that, locally, each sample's contribution to the Hessian is exactly equal to its approximation to the Hessian of its NTK-based logistic regression (i.e., the K-FAC approximation is exact in this sense).
>
> For deeper neural networks with other activation functions, the rank of the Hessian update would be at most equal to the depth of the neural network. However, we expect that a rescaling based on the rank-1 Hessian update corresponding to the NTK approximation would be a good approximation to the leave-one-out Hessian.
>
> All of these updates are rank-1, allowing for fast evaluation using the Sherman-Morrison formula and using the low-rank embedding procedure described above.
>
> **Comparison of Runtimes**
>
> | Dataset      | Training | Hessian | Inversion | Influence | Rescaling        |
> |--------------|----------|---------|-----------|-----------|------------------|
> | ESC50        | 1.8 s    | 0.056 s | 0.0005 s  | 0.051 s   | 0.0033 s (0.2%)  |
> | CatDog       | 76 s     | 4.9 s   | 0.010 s   | 4.8 s     | 0.087 s (0.1%)   |
> | AutoTruck    | 48 s     | 4.9 s   | 0.0094 s  | 4.8 s     | 0.087 s (0.2%)   |
> | DogFish      | 0.43 s   | 0.92 s  | 0.0095 s  | 0.89 s    | 0.015 s (0.7%)   |
> | Enron        | 6.7 s    | 15 s    | 0.065 s   | 15 s      | 0.095 s (0.3%)   |
> | IMDB ($n=16d$) | 20 s     | 0.92 s  | 0.0012 s  | 0.87 s    | 0.044 s (0.2%)   |

---

> > ### Comment · Area_Chair_B7Nd · 2025-08-05
> >
> > Hi Reviewer xHGk,
> >
> > Would you check the authors’ reply soon? Have they addressed your concerns?
> >
> > Best
> >
> > AC

---

> > ### Comment · Reviewer_xHGk · 2025-08-06
> >
> > The issues regarding hyperparameter sensitivity and the lack of experimental validation on large models significantly hinder the practical applicability of the RIF method. These remain critical unresolved problems. Unfortunately, the response fails to provide effective solutions or meaningful results addressing these concerns.

---

### Comment · Area_Chair_B7Nd · 2025-08-01

Hi reviewers,

Would you please check the rebuttal of the authors and comments from other reviewers, and give your thoughts about the authors’ reply? Thank you.


Best regards

AC

---

### Note · Authors · 2025-08-16

We thank the reviewers for their thoughtful and constructive feedback.

Our paper introduces rescaled influence functions (RIF) -- a simple correction to classical influence functions (IF) that significantly increases the accuracy of their predictions with a very small computational overhead.

We evaluate RIF on sample problems to provide an empirical comparison with IF and to answer fundamental open questions such as why IF captures the direction of an effect much better than it does its magnitude (Koh et al. 2019).

**Limitations:**


The main limitation raised in the reviews is our focus on convex optimizations of moderate scale.

As we note in our rebuttal, while many of the applications of data attribution are for non-convex problems and require approximate Hessian computations due to their large scale, we believe that the moderate-scale convex optimizations provide a better setting for evaluating fundamental questions such as the effect of rescaling, and we summarize our arguments below.

First, previous results focusing on the convex setting (such as the seminal works of Koh and Liang 2017, Koh et al. 2019 and Giordano et al. 2019) proved instrumental in developing techniques for the large-scale non-convex setting such as TRAK.

Moreover, our RIF method translates to the non-convex setting with a similar qualitative behavior compared to the traditional IF method. Even in the large scale setting, RIF can be estimated to a similar degree of accuracy using the same random projection techniques used to estimate IF.

Finally, the moderate-scale convex setting allows us to produce a clearer comparison between IF and RIF, since in this setting both IF and RIF can be evaluated exactly, existing analytical techniques can still be used to derive meaningful theoretical guarantees, and we avoid the "fragility" issue raised by Basu et al. 2020.


Therefore, we are confident that our findings not only answer fundamental questions about the behavior of influence functions in the convex setting, but also lay the groundwork for better data attribution in larger non-convex settings.

---

### Decision · Program_Chairs · 2025-09-17

**Decision:**

Accept (poster)

**Comment:**

This work handles a critical limitation of traditional influence functions, namely, their inaccuracy in high-dimensional or over-parameterized regimes. As pointed out by reviewers, this work well addresses the well-known limitation of IFs within modern over-parameterized models, and provides an elegant and straightforward solution with substantial potential for different applications. The authors have done extensive experiments, which show the effectiveness of this work. Due to the contributions of this work, 4 out of the 5 reviewers recommend acceptance. Considering the importance of the addressed problem, and the effectiveness of the method, AC recommends acceptance, but urges the authors to improve the final version following reviewers' comments.